# SELF-SUPERVISED LEARNING IS MORE ROBUST TO DATASET IMBALANCE

**Hong Liu**
Stanford University
`hliu99@stanford.edu`

**Jeff Z. HaoChen**
Stanford University
`jhaochen@stanford.edu`

**Adrien Gaidon**
Toyota Research Institute
`adrien.gaidon@tri.global`

**Tengyu Ma**
Stanford University
`tengyuma@stanford.edu`

## ABSTRACT

Self-supervised learning (SSL) is a scalable way to learn general visual representations since it learns without labels. However, large-scale unlabeled datasets in the wild often have long-tailed label distributions, where we know little about the behavior of SSL. In this work, we systematically investigate self-supervised learning under dataset imbalance. First, we find out via extensive experiments that off-the-shelf self-supervised representations are already more robust to class imbalance than supervised representations. The performance gap between balanced and imbalanced pre-training with SSL is significantly smaller than the gap with supervised learning, across sample sizes, for both in-domain and, especially, out-of-domain evaluation. Second, towards understanding the robustness of SSL, we hypothesize that SSL learns richer features from frequent data: it may learn label-irrelevant-but-transferable features that help classify the rare classes and downstream tasks. In contrast, supervised learning has no incentive to learn features irrelevant to the labels from frequent examples. We validate this hypothesis with semi-synthetic experiments and theoretical analyses on a simplified setting. Third, inspired by the theoretical insights, we devise a re-weighted regularization technique that consistently improves the SSL representation quality on imbalanced datasets with several evaluation criteria, closing the small gap between balanced and imbalanced datasets with the same number of examples.

## 1 INTRODUCTION

Self-supervised learning (SSL) is an important paradigm of machine learning, because it can leverage the availability of large-scale unlabeled datasets to learn representations for a wide range of downstream tasks and datasets (He et al., 2020; Chen et al., 2020; Grill et al., 2020; Caron et al., 2020; Chen & He, 2021). Current SSL algorithms are mostly trained on curated, balanced datasets, but large-scale unlabeled datasets in the wild are inevitably imbalanced with a long-tailed label distribution (Reed, 2001; Liu et al., 2019). Curating a class-balanced unlabeled dataset requires the knowledge of labels, which defeats the purpose of leveraging unlabeled data by SSL.

The behavior of SSL algorithms under dataset imbalance remains largely underexplored in the literature, but extensive studies do not bode well for supervised learning (SL) with imbalanced datasets. The performance of vanilla supervised methods degrades significantly on class-imbalanced datasets (Cui et al., 2019; Cao et al., 2019; Buda et al., 2018), posing challenges to practical applications such as instance segmentation (Tang et al., 2020) and depth estimation (Yang et al., 2021). Many recent works address this issue with various regularization and re-weighting/re-sampling techniques (Ando & Huang, 2017; Wang et al., 2017b; Jamal et al., 2020; Cui et al., 2019; Cao et al., 2019; 2021; Tian et al., 2020; Hong et al., 2021; Wang et al., 2021).

In this work, we systematically investigate the representation quality of SSL algorithms under class imbalance. Perhaps surprisingly, we find out that off-the-shelf SSL representations are already more robust to dataset imbalance than the representations learned by supervised pre-training. We evaluate

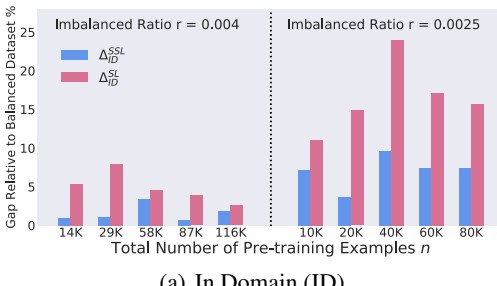 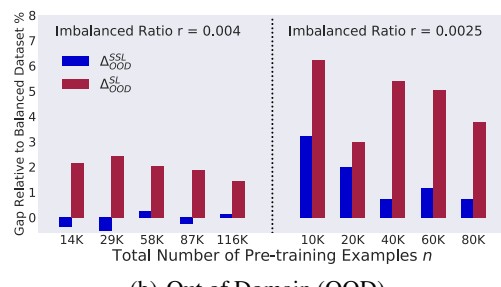

(a) In Domain (ID).                    (b) Out of Domain (OOD).

Figure 1: **Relative performance gap** (lower is better) between imbalanced and balanced representation learning. The gap is much smaller for self-supervised (MoCo v2) representations ($\Delta^{SL}$ in blue) vs. supervised ones ($\Delta^{SL}$ in red) on long-tailed ImageNet with various number of examples $n$, across both ID (a) and OOD (b) evaluations. See Equation (1) for the precise definition of the relative performance gap and and Figure 2 for the absolute performance.

the representation quality by linear probe on in-domain (ID) data and finetuning on out-of-domain (OOD) data. We compare the robustness of SL and SSL representations by computing the gap between the performance of the representations pre-trained on balanced and imbalanced datasets of the same sizes. We observe that the balance-imbalance gap for SSL is much smaller than SL, under a variety of configurations with varying dataset sizes and imbalance ratios and with both ID and OOD evaluations (see Figure 1 and Section 2 for more details). This robustness holds even with the same number of samples for SL and SSL, although SSL does not require labels and hence can be more easily applied to larger datasets than SL.

Why is SSL more robust to dataset imbalance? We identify the following underlying cause to answer this fundamental question: SSL learns richer features from the frequent classes than SL does. These features may help classify the rare classes under ID evaluation and are transferable to the downstream tasks under OOD evaluation. For simplicity, consider the situation where rare classes have so limited data that both SL and SSL models overfit to the rare data. In this case, it is important for the models to learn diverse features from the frequent classes which can help classify the rare classes. Supervised learning is only incentivized to learn those features relevant to predicting frequent classes and may ignore other features. In contrast, SSL may learn the structures within the frequent classes better—because it is not supervised or incentivized by any labels, it can learn not only the label-relevant features but also other interesting features capturing the intrinsic properties of the input distribution, which may generalize/transfer better to rare classes and downstream tasks.

We empirically validate this intuition by visualizing the features on a semi-synthetic dataset where the label-relevant features and label-irrelevant-but-transferable features are prominently seen by design (cf. Section 3.2). In addition, we construct a toy example where we can rigorously prove the difference between self-supervised and supervised features in Section 3.1.

Finally, given our theoretical insights, we take a step towards further improving SSL algorithms, closing the small gap between SSL on balanced and imbalanced datasets. We identify the generalization gap between the empirical and population pre-training losses on rare data as the key to improvements.

To this end, we design a simple algorithm that first roughly estimates the density of examples with kernel density estimation and then applies a larger sharpness-based regularization (Foret et al., 2020) to the estimated rare examples. Our algorithm consistently improves the representation quality under several evaluation protocols.

We sum up our contributions as follows. (1) We are the first to systematically investigate the robustness of self-supervised representation learning to dataset imbalance. (2) We propose and validate an explanation of this robustness of SSL, empirically and theoretically. (3) We propose a principled method to improve SSL under unknown dataset imbalance.

## 2 EXPLORING THE EFFECT OF CLASS IMBALANCE ON SSL

Dataset class imbalance can pose challenge to self-supervised learning in the wild. Without access to labels, we cannot know in advance whether a large-scale unlabeled dataset is imbalanced. We need to

study how SSL will behave under dataset imbalance to deploy SSL in the wild safely. In this section, we systematically investigate the effect of class imbalance on self-supervised representations.

## 2.1 PROBLEM FORMULATION

**Class-imbalanced pre-training datasets.** We assume the datapoints / inputs are in $\mathbb{R}^d$ and come from $C$ underlying classes. Let $x$ denote the input and $y$ denote the corresponding label. Supervised pre-training algorithms have access to the inputs and corresponding labels, whereas self-supervised pre-training only observes the inputs. Given a pre-training distribution $\mathcal{P}$ over over $\mathbb{R}^d \times [C]$, let $r$ denote the ratio of class imbalance. That is, $r$ is the ratio between the probability of the rarest class and the most frequent class: $r = \frac{\min_{j \in [C]} \mathcal{P}(y=j)}{\max_{j \in [C]} \mathcal{P}(y=j)} \leq 1$. We will construct distributions with varying imbalance ratios and use $\mathcal{P}^r$ to denote the distribution with ratio $r$. We also use $\mathcal{P}^{\text{bal}}$ for the case where $r = 1$, i.e. the dataset is balanced. Large-scale data in the wild often follow heavily long-tailed label distributions where $r$ is small. We assume that for any class $j \in [C]$, the class-conditional distribution $\mathcal{P}^r(x|y=j)$ is the same across balanced and imbalanced datasets for all $r$. The pre-training dataset $\widehat{\mathcal{P}}^r_n$ consists of $n$ i.i.d. samples from $\mathcal{P}^r$.

**Pre-trained models.** A feature extractor is a function $f_\phi : \mathbb{R}^d \to \mathbb{R}^m$ parameterized by neural network parameters $\phi$, which maps inputs to representations. A linear head is a linear function $g_\theta : \mathbb{R}^m \to \mathbb{R}^C$, which can be composed with $f_\phi$ to produce the predictions. SSL algorithms learn $\phi$ from unlabeled data. Supervised pre-training learns the feature extractor and the linear head from labeled data. We drop the head and only evaluate the quality of feature extractor $\phi$.[1]

Following the standard evaluation protocol in prior works (He et al., 2020; Chen et al., 2020), we measure the quality of learned representations on both *in-domain* and *out-of-domain* datasets with either *linear probe* or *fine-tuning*, as detailed below.

**In-domain (ID) evaluation** tests the performance of representations on the *balanced in-domain* distribution $\mathcal{P}^{\text{bal}}$ with **linear probe**. Given a feature extractor $f_\phi$ pre-trained on a pre-training dataset $\widehat{\mathcal{P}}^r_n$ with $n$ data points and imbalance ratio $r$, we train a $C$-way linear classifier $\theta$ on top of $f_\phi$ on a *balanced* dataset[2] sampled i.i.d. from $\mathcal{P}^{\text{bal}}$. We evaluate the representation quality with the top-1 accuracy of the learned linear head on $\mathcal{P}^{\text{bal}}$. We denote the ID accuracy of supervised pre-trained representations by $A^{\text{SL}}_{\text{ID}}(n,r)$. Note that $A^{\text{SL}}_{\text{ID}}(n,1)$ stands for the result with balanced pre-training dataset. For SSL representations, we denote the accuracy by $A^{\text{SSL}}_{\text{ID}}(n,r)$.

**Out-of-domain (OOD) evaluation** tests the performance of representations by *fine-tuning* the feature extractor and the head on a (or multiple) *downstream* target distribution $\mathcal{P}_t$. Starting from a feature extractor $f_\phi$ (pre-trained on a dataset of size $n$ and imbalance ratio $r$) and a randomly initialized classifier $\theta$, we fine-tune $\phi$ and $\theta$ on the target dataset $\widehat{\mathcal{P}}_t$, and evaluate the representation quality by the expected top-1 accuracy on $\mathcal{P}_t$. We use $A^{\text{SL}}_{\text{OOD}}(n,r)$ and $A^{\text{SSL}}_{\text{OOD}}(n,r)$ to denote the resulting accuracies of supervised and self-supervised representations, respectively.

**Summary of varying factors.** We aim to study the effect of class imbalance to feature qualities on a diverse set of configurations with the following varying factors: (1) the number of examples in pre-training $n$, (2) the imbalance ratio of the pre-training dataset $r$, (3) ID or OOD evaluation, and (4) self-supervised learning algorithms: MoCo v2 (He et al., 2020), or SimSiam (Chen & He, 2021).

## 2.2 EXPERIMENTAL SETUP

**Datasets.** We pre-train the representations on variants of ImageNet (Russakovsky et al., 2015) or CIFAR-10 (Krizhevsky & Hinton, 2009) with a wide range of numbers of examples and ratios of imbalance. Following Liu et al. (2019), we consider exponential and Pareto distributions, which closely simulate the natural long-tailed distributions. We consider imbalance ratio in $\{1, 0.004, 0.0025\}$ for ImageNet and $\{1, 0.1, 0.01\}$ for CIFAR-10. For each imbalance ratio, we further downsample the dataset with a sampling ratio in $\{0.75, 0.5, 0.25, 0.125\}$ to form datasets with varying sizes. We

---

[1]It is well-known that the composition of the head and features learned from supervised learning is more sensitive to imbalanced dataset than feature extractor $\phi$ itself (Cao et al., 2019; Kang et al., 2020). Please also see Table 3 in Appendix C for a comparison between CRT (Kang et al., 2020) and Supervised.

[2]We essentially use the largest balanced labeled ID dataset for this evaluation, which oftentimes means the entire curated training dataset, such as CIFAR-10 with 50,000 examples and ImageNet with 1,281,167 examples.

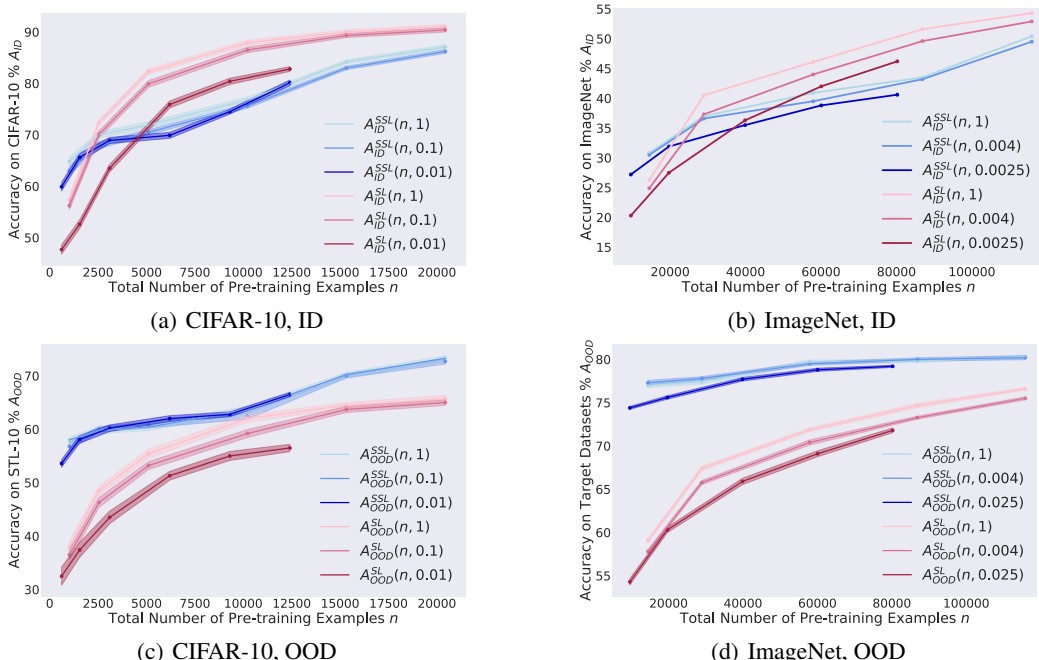

Figure 2: **Representation quality on balanced and imbalanced datasets.** Left: CIFAR-10, SL vs. SSL (SimSiam); Right: ImageNet, SL vs. SSL (MoCo v2). For both ID and OOD, the gap between balanced and imbalanced datasets with the same $n$ is larger for supervised learning. The accuracy of supervised representations is better with reasonably large $n$ in ID evaluation, while self-supervised representations perform better in OOD evaluation.[4]

fix the variant of the dataset when comparing different algorithms. For ID evaluation, we use the original CIFAR-10 or ImageNet training set for the training phase of linear probe and use the original validation set for the final evaluation. For OOD evaluation of representations learned on CIFAR-10, we use STL-10 (Coates et al., 2011) as the target /downstream dataset. For OOD evaluation of representations learned on ImageNet, we fine-tune the pre-trained feature extractors on CUB-200 (Wah et al., 2011), Stanford Cars (Krause et al., 2013), Oxford Pets (Parkhi et al., 2012), and Aircrafts (Maji et al., 2013), and measure the representation quality with average accuracy on the downstream tasks.

**Models.** We use ResNet-18 on CIFAR-10 and ResNet-50 on ImageNet as backbones. For supervised pre-training, we follow the standard protocol of He et al. (2016) and Kang et al. (2020). For self-supervised pre-training, we consider MoCo v2 (He et al., 2020) and SimSiam (Chen & He, 2021). We run each evaluation experiment with 3 seeds and report the average and standard deviation in the figures. Further implementation details and additional results are deferred to Section A.

### 2.3 RESULTS: SELF-SUPERVISED LEARNING IS MORE ROBUST THAN SUPERVISED LEARNING TO DATASET IMBALANCE

In Figure 2, we plot the results of ID and OOD evaluations, respectively. For both ID and OOD evaluations, the gap between SSL representations learned on balanced and imbalanced datasets with the same number of pre-training examples, i.e., $A^{\text{SSL}}(n, 1) - A^{\text{SSL}}(n, r)$, is smaller than the gap of supervised representations, i.e., $A^{\text{SL}}(n, 1) - A^{\text{SL}}(n, r)$, consistently in all configurations. Furthermore, we compute the relative accuracy gap to balanced dataset $\Delta^{\text{SSL}}(n, r) \triangleq (A^{\text{SSL}}(n, 1) - A^{\text{SSL}}(n, r))/A^{\text{SSL}}(n, 1)$ in Figure 1. We observe that with the same number of pre-training examples, the relative gap of SSL representations between balanced and imbalanced datasets is smaller than that of SL representations across the board,

$$\Delta^{\text{SSL}}(n, r) \triangleq \frac{A^{\text{SSL}}(n, 1) - A^{\text{SSL}}(n, r)}{A^{\text{SSL}}(n, 1)} \ll \Delta^{\text{SL}}(n, r) \triangleq \frac{A^{\text{SL}}(n, 1) - A^{\text{SL}}(n, r)}{A^{\text{SL}}(n, 1)}. \quad (1)$$

---

[4]The maximum $n$ is smaller for extreme imbalance. The standard deviation comes only from the randomness of evaluation. We do not include the stddev for ImageNet ID due to limitation of computation resources.

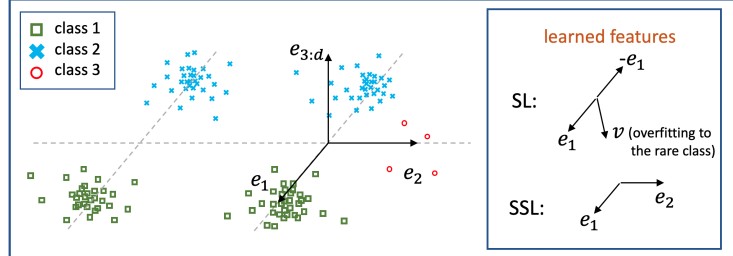

Figure 3: **Explaining SSL's robustness in a toy setting.** $e_1$ and $e_2$ are two orthogonal directions in the $d$-dimensional Euclidean space that decides the labels, and $e_{3:d}$ represents the other $d - 2$ dimensions. Classes 1 and 2 are frequent classes and the third class is rare. To classify the three classes, the representations need to contain both $e_1$ and $e_2$ directions. Supervised learning learns direction $e_1$ from the frequent classes (which is necessary and sufficient to identify classes 1 and 2) and some overfitting direction $v$ from the rare class which has insufficient data. Note that $v$ might be mostly in the $e_{3:d}$ directions due to overfitting. In contrast, SSL learns both $e_1$ and $e_2$ directions from the frequent classes because they capture the intrinsic structures of the inputs (e.g., $e_1$ and $e_2$ are the directions with the largest variances), even though $e_2$ does not help distinguish the frequent classes. The direction $e_2$ learned from frequent data by SSL can help classify the rare class.

Also note that comparing the robustness with the same number of data is actually in favor of SL, because SSL is more easily applied to larger datasets without the need of collecting labels.

**ID vs. OOD.** As shown in Figure 2, we observe that representations from supervised pre-training perform better than self-supervised pre-training in ID evaluation with reasonably large $n$, while self-supervised pre-training is better in OOD evaluation. This phenomenon is orthogonal to our observation that SSL is more robust to dataset imbalance, and is consistent with recent works (e.g., Chen et al. (2020); He et al. (2020)) which also observed that SSL performs slightly worse than supervised learning on balanced ID evaluation but better on OOD tasks.

## 3 ANALYSIS

We have found out with extensive experiments that self-supervised representations are more robust to class imbalance than supervised representations. A natural and fundamental question arises: where does the robustness stem from? In this section, we propose a possible reason and justify it with theoretical and empirical analyses.

**SSL learns richer features from frequent data that are transferable to rare data.** The rare classes of the imbalanced dataset can contain only a few examples, making it hard to learn proper features to classify the rare classes. In this case, one may want to resort to the features learned from the frequent classes for help. However, due to the supervised nature of classification tasks, the supervised model mainly learns the features that help classify the frequent classes and may neglect other features which can transfer to the rare classes and potentially the downstream tasks. Partly because of this, Jamal et al. (2020) explicitly encourage the model to learn features transferable from the frequent to the rare classes with meta-learning. In contrast, in self-supervised learning, without the bias or incentive from the labels, the models can learn richer features that capture the intrinsic structures of the inputs—both features useful for classifying the frequent classes and features transferable to the rare classes.

### 3.1 RIGOROUS ANALYSIS ON A TOY SETTING

To justify the above conjecture, we instantiate supervised and self-supervised learning in a setting where the features helpful to classify the frequent classes and features transferable to the rare classes can be clearly separated. In this case, we prove that self-supervised learning learns better features than supervised learning.

**Data distribution.** Let $e_1, e_2$ be two orthogonal unit-norm vectors in the $d$-dimensional Euclidean space. Consider the following pre-training distribution $\mathcal{P}$ of a 3-way classification problem, where the class label $y \in [3]$. The input $x$ is generated as follows. Let $\tau > 0$ and $\rho > 0$ be hyperparameters of the distribution. First sample $q$ uniformly from $\{0, 1\}$ and $\xi \sim \mathcal{N}(0, I)$ from Gaussian distribution. For the first class ($y = 1$), set $x = e_1 - q\tau e_2 + \rho\xi$. For the second class ($y = 2$), set $x =$

$-e_1 - q\tau e_2 + \rho\xi$. For the third class ($y = 3$), set $x = e_2 + \rho\xi$. Classes 1 and 2 are frequent classes, while class 3 is the rare class, i.e., $\frac{\mathcal{P}(y=3)}{\mathcal{P}(y=1)}, \frac{\mathcal{P}(y=3)}{\mathcal{P}(y=2)} = o(1)$. See Figure 3 for an illustration of this data distribution. In this case, both $e_1$ and $e_2$ are features from the frequent classes 1 and 2. However, only $e_1$ helps classify the frequent classes and only $e_2$ can be transferred to the rare classes.

**Algorithm formulations.** For supervised learning, we train a two-layer linear network $f_{W_1,W_2}(x) \triangleq W_2 W_1 x$ with weight matrices $W_1 \in \mathbb{R}^{m \times d}$ and $W_2 \in \mathbb{R}^{3 \times m}$ for some $m \geq 3$, and then use the first layer $W_{\text{SL}} = W_1$ as the feature for downstream tasks. Given a linearly separable labeled dataset, we learn such a network with minimal norm $\|W_1^\top W_1\|_F^2 + \|W_2^\top W_2\|_F^2$ subject to the margin constraint $f_{W_1,W_2}(x)_y \geq f_{W_1,W_2}(x)_{y'} + 1$ for all data $(x, y)$ in the dataset and $y' \neq y$.[5] For self-supervised learned, similar to SimSiam (Chen et al., 2020), we construct positive pairs $(x + \xi, x + \xi')$ where $x$ is from the empirical dataset, $\xi$ and $\xi'$ are independent random perturbations. We learn a matrix $W_{\text{SSL}} \in \mathbb{R}^{m \times d}$ which minimizes $-\hat{\mathbb{E}}[(W(x + \xi))^T(W(x + \xi'))] + \frac{1}{2}\|W^\top W\|_F^2$, where the expectation $\hat{\mathbb{E}}$ is over the empirical dataset and the randomness of $\xi$ and $\xi'$. The regularization term $\frac{1}{2}\|W^\top W\|_F^2$ is introduced only to make the learned features more mathematically tractable. We use $W_{\text{SSL}}x$ as the feature of data $x$ in the downstream task.

**Main intuitions.** We compare the features learned by SSL and supervised learning on an imbalanced dataset that contains an abundant (poly in $d$) number of data from the frequent classes but only a small (sublinear in $d$) number of data from the rare class. The key intuition behind our analysis is that supervised learning learns only the $e_1$ direction (which helps classify class 1 vs. class 2) and some random direction that overfits to the rare class. In contrast, self-supervised learning learns both $e_1$ and $e_2$ directions from the frequent classes. Since how well the feature helps classify the rare class (in ID evaluation) depends on how much it correlates with the $e_2$ direction, SSL provably learns features that help classify the rare class, while supervised learning fails. This intuition is formalized by the following theorem.

**Theorem 3.1.** *Let $n_1, n_2, n_3$ be the number of data from the three classes respectively. Let $\rho = d^{-\frac{1}{5}}$ and $\tau = d^{\frac{1}{5}}$ in the data generative model. For $n_1, n_2 = \Theta(\text{poly}(d))$ and $n_3 \leq d^{\frac{1}{5}}$, with probability at least $1 - O(e^{-d^{\frac{1}{10}}})$, the following statements hold for any feature dimension $m \geq 3$:*

- *Let $W_{\text{SL}} = [w_1, w_2, \cdots, w_m]^\top$ be the feature learned by SL, then $\sum_{i=1}^{m}\langle e_2, w_i\rangle^2 \leq O(d^{-\frac{1}{10}})$.*
- *Let $W_{\text{SSL}} = [\tilde{w}_1, \tilde{w}_2, \cdots, \tilde{w}_m]^\top$ be the feature learned by SSL, then $\|\Pi e_2\|_2 \geq 1 - O(d^{-\frac{1}{5}})$, where $\Pi$ projects $e_2$ onto the row span of $W_{\text{SSL}}$.*

Supervised learning results in features $W_{\text{SL}}$ whose rows have small correlation with the transferable feature $e_2$, indicating that supervised learning only learns features for classifying the frequent classes and ignore the transferable features. In contrast, self-supervised learning recovers $e_2$ well, even though $e_2$ is not relevant to classifying the frequent classes. The proofs are deferred to Section E.

## 3.2 Illustrative Semi-synthetic Experiments

In the previous subsection, we have shown that self-supervised learning provably learns label-irrelevant-but-transferable features from the frequent classes which can help classify the rare class in the toy case, while supervised learning mainly focuses on the label-relevant features. However, in real-world datasets, it is intractable to distinguish the two groups of features. To amplify this effect in a real-world dataset and highlight the insight of the theoretical analysis, we design a semi-synthetic experiment on SimCLR (Chen et al., 2020) to validate our conclusion.

**Dataset.** In the theoretical analysis above, the frequent classes contain both features related to the classification of frequent classes and features transferable to the the rare classes. Similarly, we consider an imbalanced pre-training dataset with two groups of features modified from CIFAR-10 as shown in Figure 4 (Left). We construct classes 1-5 as the frequent classes, where each class contains 5000 examples. Classes 6-10 are the rare classes, where each class has 10 examples. In this case, the ratio of imbalance $r = 0.002$. Each image from classes 1-5 consists of a left half and a right half. The left half of an example is from classes 1-5 of the original CIFAR-10 and corresponds to the label of that example. The right half is from a random image of CIFAR-10, which is label-irrelevant. In

---

[5]Previous work shows that deep linear networks trained with gradient descent using logistic loss converge to this min norm solution in direction (Ji & Telgarsky, 2018).

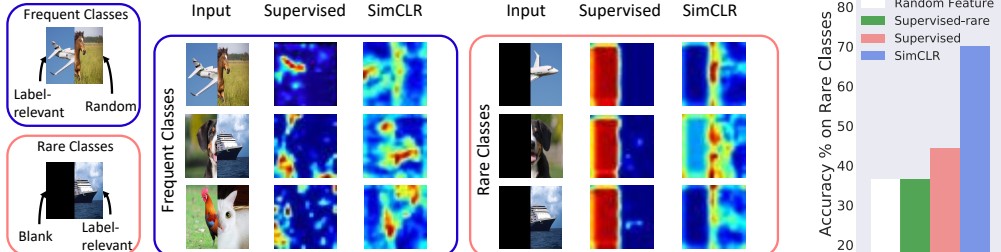

Figure 4: **Visualization of SSL features in semi-synthetic settings. Left**: The right halves of the rare examples decide the labels, while the left are blank. The left halves of the frequent examples decide the labels, while the right halves are random half images, which contain label-irrelevant-but-transferable features. **Middle**: Visualization of features with Grad-CAM (Selvaraju et al., 2017). SimCLR learns features from both left and right sides, whereas SL mainly learns label-relevant features from the left and ignore label-irrelevant features on the right. **Right**: Accuracies evaluated on rare classes. The head classifiers are trained on 25000 examples from the 5 rare classes. SimCLR learns much better features for rare classes than SL. We include random feature (feature extractor with random weights) and supervised-rare (model trained with only the rare examples) for references.

contrast, the left half of an example from classes 6-10 is blank, whereas the right half is label-relevant and from classes 6-10 of the original CIFAR-10. In this setting, features from the left halves of the images are correlated to the classification of the frequent classes, while features from the right halves are label-irrelevant for the frequent classes, but can help classify the rare classes. Note that features from the right halves cannot be directly learned from the rare classes since they have only 10 examples per class. This is consistent with the setting of Theorem 3.1.

**Pre-training.** We pre-train the representations on the semi-synthetic imbalanced dataset. For supervised learning, we use ResNet-50 on this 10-way classification task. For self-supervised learning, we use SimCLR with ResNet-50. To avoid confusing the left and right parts, we disable the random horizontal flip in the data augmentation. After pre-training, we fix the representations and train a linear classifier on top of the representations with balanced data from the 5 rare classes (25000 examples in total) to test if the model learns proper features for the rare classes during pre-training. In Figure 4 (Right), we test the classifier on the rare classes. In Figure 4 (Middle), we further visualize the Grad-CAM (Selvaraju et al., 2017) of the representations on the held-out set[6].

**Results.** As a sanity check, we first pre-train a supervised model with only the 50 rare examples and train the linear head classifier with 25000 examples from the rare classes (5-way classification) to see if the model can learn proper features for the rare classes with only rare examples (Supervised-rare in Figure 4 (Right)). As expected, the accuracy is $36.5\%$, which is almost the same as randomly initialized representations with trained head classifier, indicating that the model cannot learn the features for the rare classes with only rare examples due to the limited number of examples. We then compare supervised learning with self-supervised learning on the whole semi-synthetic dataset. In Figure 4 (Right), self-supervised representations perform much better than supervised representations on the rare classes ($70.1\%$ vs $44.3\%$). We further visualize the activation maps of representations with Grad-CAM. Supervised learning mostly activate the left halves of the examples for both frequent and rare classes, indicating that it mainly learn features on the left. In sharp contrast, self-supervised learning activates the whole image on the frequent examples and the right part on the rare examples, indicating that it learns features from both parts.

## 4  IMPROVING SSL ON IMBALANCED DATASETS WITH REGULARIZATION

In this section, we aim to further improve the performance of SSL to close the gap between imbalanced and balanced datasets. Many prior works on imbalanced supervised learning regularize the rare classes more strongly, motivated by the observation that the rare classes suffer from more overfitting (Cao et al., 2019; 2021). Inspired by these works, we compute the generalization gaps (i.e., the differences between empirical and validation pre-training losses) on frequent and rare classes for the step-

---

[6]CIFAR images are of low resolution. For visualization, we use high resolution version of the CIFAR-10 images in Figure 4 (Middle). We also provide the visualization on original CIFAR-10 images in Figure 7.

imbalance CIFAR-10 datasets (where 5 classes are frequent class with 5000 examples per class and the rest are rare with 50 examples per class). As shown in Table 1 (a), we still observe a similar phenomenon—the frequent classes have much smaller pre-training generalization gap than the rare classes (0.035 vs. 0.081), which indicates the necessity of more regularization on the rare classes.

We need a data-dependent regularizer that can have different effects on rare and frequent examples. Thus, weight decay or dropout (Srivastava et al., 2014) are not suitable. The prior work of Cao et al. (2019) regularizes the rare classes more strongly with larger margin, but it does not apply to SSL where no labels are available. Inspired by Cao et al. (2021), we adapt sharpness-aware minimization (SAM) (Foret et al., 2021) to imbalanced SSL.

**Reweighted SAM (rwSAM).** SAM improves model generalization by penalizing loss sharpness. Suppose the training loss of the representation $f_\phi$ is $\widehat{L}(\phi)$, i.e. $\widehat{L}(\phi) = \frac{1}{n} \sum_{j=1}^n \ell(x_j, \phi)$. SAM seeks parameters where the loss is uniformly low in the neighboring area,

$$\min_\phi \widehat{L}(\phi + \epsilon(\phi)), \quad \text{where} \quad \epsilon(\phi) = \arg\max_{\|\epsilon\| < \rho} \epsilon^\top \nabla_\phi \widehat{L}(\phi). \tag{2}$$

To take the weight of different examples into account, we add reweighting to the inner maximization step of SAM. Intuitively, we wish the optimization landscape to be flatter for rare examples, which is in effect regularizing the model more on rare examples. Concretely, consider the reweighted training loss associated with weight vector $w \in \mathbb{R}^n$, $\widehat{L}_w(\phi) = \frac{1}{n} \sum_{j=1}^n w_j \ell(x_j, \phi)$. The reweighted SAM objective re-weights the regularization-related terms (e.g., $\epsilon_w$) but not the training loss $\widehat{L}$:

$$\min_\phi \widehat{L}(\phi + \epsilon_w(\phi)), \quad \text{where} \quad \epsilon_w(\phi) = \arg\max_{\|\epsilon\| < \rho} \epsilon^\top \nabla_\phi \widehat{L}_w(\phi). \tag{3}$$

**Assigning Weight with Kernel Density Estimation.** The weight $w_j$ of an example $x_j$ should be inversely correlated with the frequency of the corresponding class $y_j$. However, we have no access to the labels. In order to approximate the frequency, we use kernel density estimation on top of the representations $f_\phi$. Concretely, denote by $K(\cdot, h)$ the Gaussian density with bandwidth $h$. We assign $w_i$ to be inversely correlated with the estimated density, i.e., $w_i = \left( \frac{1}{n} \sum_{j=1}^n K(f_\phi(x_i) - f_\phi(x_j), h) \right)^{-\alpha}$ where $h$ and $\alpha > 0$ are hyperparameters selected by cross validation.

### 4.1 EXPERIMENTS

We test the proposed rwSAM on CIFAR-10 with step or exponential imbalance and ImageNet-LT (Liu et al., 2019). After self-supervised pre-training on the long-tailed dataset, we evaluate the representations by (1) linear probing on the balanced in-domain dataset and (2) fine-tuning on downstream target datasets. For (1) and (2), we compare with SSL, SSL+SAM (w/o reweighting), and SSL balanced, which learns the representations on the balanced dataset *with the same number of examples*. Implementation details and additional results are deferred to Section C. Code is available at `https://github.com/Liuhong99/Imbalanced-SSL`.

**Results.** Table 1 (a) summarizes results on long tailed CIFAR-10. With both step and exponential imbalance, rwSAM improves the performance of SimSiam over $1\%$, and even surpasses the performance of SimSiam on balanced CIFAR-10 with the same number of examples. Note that compared to SimSiam, rwSAM closes the generalization gap of pre-training loss on rare examples from 0.081 to 0.066, which verifies the effect of re-weighted regularization. In Table 1 (b), we provide the result of fine-tuning on downstream tasks with representations pre-trained on ImageNet-LT. The proposed method improves the transferability of representations to downstream tasks consistently.

## 5 RELATED WORK

**Supervised Learning with Dataset Imbalance.** There exists a line of works studying supervised imbalanced classification. Ando & Huang (2017); Buda et al. (2018) proposed to re-sample the data to make the frequent and rare classes appear with equal frequency in training. Re-weighting assigns different weights for head and tail classes and eases the optimization difficulty under class imbalance (Cui et al., 2019; Tang et al., 2020; Wang et al., 2017b). Byrd & Lipton (2019); Xu et al. (2021) studied the effect of importance weighting and found out that importance weighting does not change the solution without regularization. Cao et al. (2019) studied reweighted regularization based

| (a) CIFAR, ID | $r = 0.01$, step | | | $r = 0.01$, exp |
|---|---|---|---|---|
| Method | Acc. (%) | Gap Freq. | Gap Rare | Acc. (%) |
| SimSiam | $84.3 \pm 0.2$ | 0.035 | 0.081 | $81.4 \pm 0.3$ |
| SimSiam+SAM | $84.7 \pm 0.3$ | 0.044 | 0.075 | $82.1 \pm 0.4$ |
| SimSiam, balanced | $85.8 \pm 0.2$ | 0.038 | 0.037 | $82.0 \pm 0.4$ |
| SimSiam+rwSAM | $85.6 \pm 0.4$ | 0.037 | 0.066 | $82.7 \pm 0.5$ |

| (b) ImageNet, OOD | Target dataset | | | | |
|---|---|---|---|---|---|
| Method | CUB | Cars | Aircrafts | Pets | Avg. |
| MoCo v2 | $69.9 \pm 0.7$ | $88.4 \pm 0.4$ | $82.9 \pm 0.6$ | $80.1 \pm 0.6$ | 80.3 |
| MoCo v2+SAM | $69.9 \pm 0.5$ | $88.8 \pm 0.5$ | $83.4 \pm 0.4$ | $81.5 \pm 0.8$ | 80.9 |
| MoCo v2, balanced | $69.8 \pm 0.5$ | $88.6 \pm 0.4$ | $82.7 \pm 0.5$ | $80.0 \pm 0.4$ | 80.2 |
| MoCo v2+rwSAM | $70.3 \pm 0.7$ | $88.7 \pm 0.3$ | $84.9 \pm 0.6$ | $81.7 \pm 0.4$ | 81.4 |
| SimSiam | $70.0 \pm 0.3$ | $87.0 \pm 0.6$ | $81.5 \pm 0.7$ | $83.8 \pm 0.5$ | 80.6 |
| SimSiam, balanced | $70.5 \pm 0.8$ | $87.9 \pm 0.7$ | $81.8 \pm 0.7$ | $82.7 \pm 0.4$ | 80.7 |
| SimSiam+rwSAM | $70.7 \pm 0.8$ | $88.4 \pm 0.6$ | $82.6 \pm 0.6$ | $84.0 \pm 0.4$ | 81.4 |

Table 1: **Results of the proposed rwSAM.** (a) Results on CIFAR-10-LT with linear probe and ID evaluation. SimSiam+rwSAM on imbalanced datasets performs even better than SimSiam on balanced datasets with the same number of examples. Note that rwSAM closes the generalization gap on the rare examples (0.081 vs. 0.066). (b) Results on ImageNet-LT with fine-tuning and OOD evaluation. rwSAM improves the performance of MoCo v2 and SimSiam on the target datasets.

on classifier margin, but these techniques are limited to supervised imbalanced recognition. Cao et al. (2021) proposed to regularize the local curvature of loss on imbalanced and noisy datasets. Recent works also designed specific losses or training pipelines for imbalanced recognition (Jamal et al., 2020; Hong et al., 2021; Wang et al., 2021; Zhang et al., 2021).

Several works also studied the supervised representations under dataset imbalance. Kang et al. (2020); Wang et al. (2020) found out that the representations of supervised learning perform better than the classifier itself with class imbalance. Yang & Xu (2020) studied the effect of self-training and self-supervised pre-training on supervised imbalanced recognition classifiers. In contrast, the focus of our paper is the effect of class imbalance on *self-supervised representations*.

**Self-supervised Learning.** Recent works on self-supervised learning successfully learn representations that approach the supervised baseline on ImageNet and various downstream tasks. Contrastive learning methods attract positive pairs and drive apart negative pairs (He et al., 2020; Chen et al., 2020). Siamese networks predict the output of the other branch, and use stop-gradient to avoid collapsing (Grill et al., 2020; Chen & He, 2021). Clustering methods learn representations by performing clustering on the representations and improve the representations with cluster index (Caron et al., 2020). Cole et al. (2021) investigated the effect of data quantity and task granularity on self-supervised representations. Goyal et al. (2021) studied self-supervised methods on large scale datasets in the wild. Kotar et al. (2021) studied whether dataset imbalance can have a significant impact on contrastive learning representations. Several works have also theoretically studied the success of self-supervised learning (Arora et al., 2019; Lee et al., 2020b; HaoChen et al., 2021; Wei et al., 2021).

# 6 CONCLUSION

Our paper is the first to study the problem of robustness to imbalanced training of self-supervised representations. We discover that self-supervised representations are more robust to class imbalance than supervised representations and explore the underlying cause of this phenomenon. Our experiments mainly focus on vision datasets. Future works can study the effect of dataset imbalance on NLP datasets, where self-supervised pre-training is a dominant approach. We hope our study can inspire analysis of self-supervised learning in broader environments in the wild such as domain shift, and provide insights for the design of future unsupervised learning methods.

## ACKNOWLEDGEMENTS

We thank Colin Wei, Margalit Glasgow, and Shibani Santurkar for helpful discussions. TM acknowledges support of Google Faculty Award, NSF IIS 2045685, the Sloan Fellowship, and JD.com. Toyota Research Institute provided funds to support this work.

## REPRODUCIBILITY STATEMENT

To ensure reproducibility, we describe the implementation details of the algorithms and the construction of the datasets in Section A.1 and Section C. The code of the experiments in Section 2, Section 3.2 and Section 4 is provided in the supplementary material. We describe the setting and data assumptions of the toy case in Section 3.1 and provide the proof in Section E.

## ETHICS STATEMENT

Our paper studies the problem of robustness to imbalanced training of self-supervised representations. This setting is important to AI Ethics, as large real-world datasets tend to be imbalanced in practice, for instance including less examples from under-represented minorities. Furthermore, pre-training is a standard practice in deep learning, especially for quickly adapting models to new domains, which corresponds to our OOD evaluation scenario.

Our experiments and theoretical analysis show that SSL is more robust than supervised pre-training, especially in the OOD scenario. As supervised learning is still the de facto standard for pre-training, our work should have a wide impact, encouraging practitioners to use SSL for pre-training instead, or at least consider evaluating the impact of imbalanced pre-training on their downstream task.

We also remark that the paper does not imply at all that the algorithms proposed or studied can guarantee any form of fairness, and they in fact should still suffer from biases. The paper should be considered as a step towards studying the important technical issue of dataset imbalance, which is related to the fairness or biases questions.

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

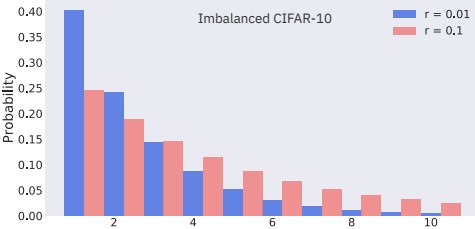 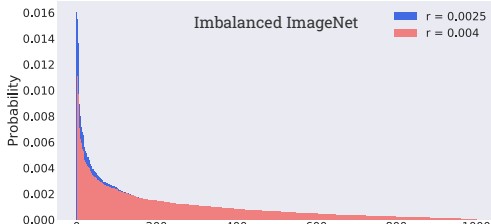

Figure 5: **Visualization of the label distributions.** We visualize the label distributions of the imbalanced CIFAR-10 and ImageNet. We consider two imbalance ratios $r$ for each dataset.Imbalanced CIFAR-10 follows the exponential distribution, while imbalanced ImageNet follows Pareto distribution.

## A    DETAILS OF SECTION 2

### A.1    IMPLEMENTATION DETAILS

**Generating Pre-training Datasets.** CIFAR-10 (Krizhevsky & Hinton, 2009) contains 10 classes with 5000 examples per class. We use exponential imbalance, i.e. for class $c$, the number of examples is $5000 \times e^{\beta(c-1)}$. we consider imbalance ratio $r \in \{0.1, 0.01\}$, i.e. the number of examples belonging to the rarest class is 500 or 50. The total $n_s$ is therefore 20431 or 12406. ImageNet-LT is constructed by Liu et al. (2019), which follows the Pareto distribution with the power value 6. The number of examples from the rarest class is 5. We construct a long tailed ImageNet following the Pareto distribution with more imbalance, where the number of examples from the rarest class is 3. The total number of examples $n_s$ is 115846 and 80218 respectively. For each ratio of imbalance, we further downsample the dataset with the sampling ratio in $\{0.75, 0.5, 0.25, 0.125\}$ to formulate different number of examples. To compare with the balanced setting fairly, we also sample balanced versions of datasets with the same number of examples. Note that each variant of the dataset is fixed after construction for all algorithms. See the visualization of label distributions of dataset variants in Figure 5.

**Training Procedure.** For supervised pre-training, we follow the standard protocol of He et al. (2016) and Kang et al. (2020). On the standard ImageNet-LT, we train the models for 90 epochs with step learning rate decay. For down-sampled variants, the training epochs are selected with cross validation. Fo self-supervised learning, the initial learning rate on the standard ImageNet-LT is set to 0.025 with batch-size 256. We train the model for 300 epochs on the standard ImageNet-LT and adopt cosine learning rate decay following (He et al., 2020; Chen & He, 2021). We train the models for more epochs on the down sampled variants to ensure the same number of total iterations. The code on CIFAR-10 LT is adapted from `https://github.com/Reza-Safdari/SimSiam-91.9-top1-acc-on-CIFAR10`.

**Evavluation.** For in-domain evaluation (ID), we first train the the representations on the aforementioned dataset variants, and then train the linear head classifier on the *full balanced* CIFAR10 or ImageNet. We set the initial learning rate to 30 when training the linear head with batch-size 4096 and train for 100 epochs in total. For in-domain out-of-domain evaluation (OOD) on ImageNet, we first train the the representations on the aforementioned dataset variants, and then fine-tune the model to CUB-200 (Wah et al., 2011), Stanford Cars (Krause et al., 2013), Oxford Pets (Parkhi et al., 2012), and Aircrafts (Maji et al., 2013). The number of examples of these target datasets ranges from 2k to 10k, which is a reasonable scale as the number of examples of the pre-training dataset variants ranges from 10k to 110k. The representation quality is evaluated with the average performance on the four tasks. We set the initial learning rate to 0.1 in fine-tuning train for 150 epochs in total. For in-domain out-of-domain evaluation (OOD) on CIFAR-10, we use STL-10 as the downstream target tasks and perform linear probe.

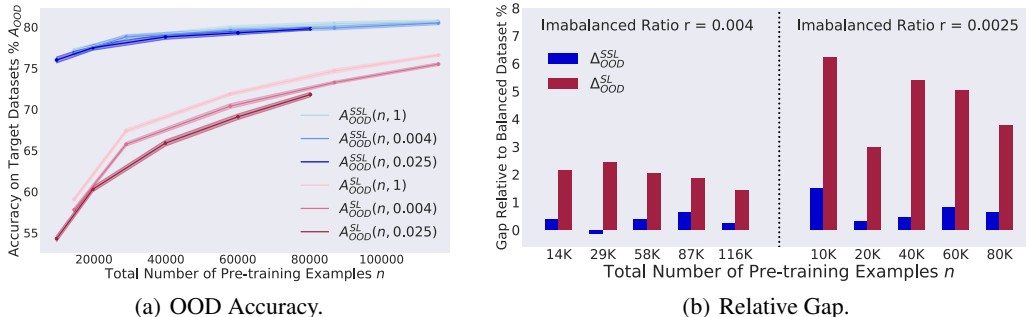

(a) OOD Accuracy.  (b) Relative Gap.

Figure 6: **OOD Results of SimSiam on ImageNet.** SimSiam also demonstrates more robustness to class imbalance compared to supervised learning. The relative gap to balanced dataset is much smaller than supervised learning across different imbalance ratios.

Table 2: Numbers in Figure 2 and Figure 6.

| Imbalanced Ratio $r$ | $r = 1$, balanced | | | | | $r = 0.004$ | | | | | $r = 0.0025$ | | | | |
|---|---|---|---|---|---|---|---|---|---|---|---|---|---|---|---|
| Data Quantity $n$ | 116K | 87K | 58K | 29K | 14K | 116K | 87K | 58K | 29K | 14K | 80K | 60K | 40K | 20K | 10K |
| MoCo V2, ID | 50.4 | 43.5 | 40.9 | 37.0 | 30.8 | 49.5 | 43.2 | 39.5 | 36.6 | 30.5 | 40.6 | 38.8 | 35.5 | 31.9 | 27.2 |
| MoCo V2, OOD | 80.3 | 79.8 | 79.7 | 77.4 | 77.0 | 80.2 | 80.1 | 79.5 | 77.8 | 77.3 | 79.2 | 78.8 | 77.7 | 75.6 | 74.4 |
| Supervised, ID | 54.3 | 51.6 | 46.1 | 40.5 | 26.3 | 52.9 | 49.6 | 44.0 | 37.3 | 24.9 | 46.1 | 42.0 | 36.3 | 27.5 | 20.3 |
| Supervised, OOD | 76.6 | 74.7 | 71.9 | 67.4 | 59.1 | 75.5 | 73.3 | 70.4 | 65.8 | 57.8 | 71.8 | 69.1 | 65.9 | 60.3 | 54.3 |
| SimSiam, OOD | 80.7 | 80.4 | 79.9 | 78.7 | 77.2 | 80.6 | 79.9 | 79.6 | 78.8 | 76.9 | 79.8 | 79.3 | 78.8 | 77.5 | 76.0 |

## A.2 ADDITIONAL RESULTS

To validate the phenomenon observed in Section 2 is consistent for different self-supervised learning algorithms, we provide the OOD evaluation results of SimSiam trained on ImageNet variants and relative performance gap with balanced datasets in Figure 6. SimSiam representations are also less sensitive to class imbalance than supervised representations.

We also provide the numbers of Figure 2 and Figure 6 in Table 2.

## B DETAILS OF SECTION 3.2

We first generate the balanced semi-synthetic dataset with 5000 examples per class. The left halves of images from classes 1-5 correspond to the labels, while the right halves are random. The left halves of images from class 6-10 are blank, whereas the right halves correspond to the labels. We then generate the imbalanced dataset, which consists of the 5000 examples per class from classes 1-5 (frequent classes), and 10 examples per class from classes 6-10 (rare classes). We use Grad-CAM implementation based on https://github.com/meliketoy/gradcam.pytorch and SimCLR implementation from https://github.com/leftthomas/SimCLR. We provide examples and Grad-CAM of the semi-synthetic datasets in Figure 7.

## C DETAILS OF SECTION 4

### C.1 IMPLEMENTATION DETAILS

We use the same implementation as Section 2 for supervised and self-supervised learning baselines. We implement sharpness-aware minimization following (Foret et al., 2021). In each step of update, we first compute the reweighted loss $\widehat{L}_w(\phi) = \frac{1}{n}\sum_{j=1}^{n} w_j \ell(x_j, \phi)$ and compute its gradient w.r.t. $\phi$, i.e. $\nabla_\phi \widehat{L}_w(\phi)$. Then we can compute $\epsilon(\phi)$ as $\epsilon(\phi) = \rho \mathrm{sgn}(\nabla_\phi \widehat{L}_w(\phi)) \left|\nabla_\phi \widehat{L}_w(\phi)\right|^{q-1} / \left(\|\nabla_\phi \widehat{L}_w(\phi)\|_q^q\right)^{1/p}$, where $\frac{1}{p} + \frac{1}{q} = 1$. Finally, we update the

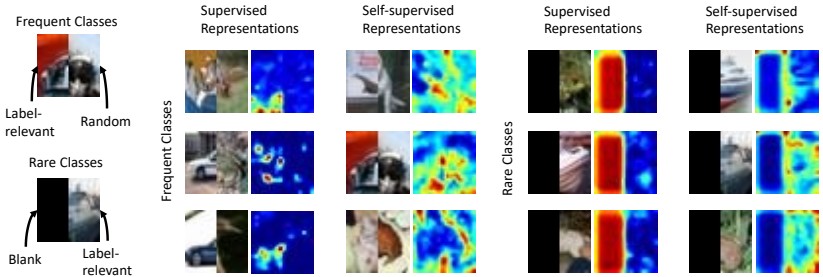

Figure 7: **Examples of the semi-synthetic Datasets and Grad-CAM visualizations.** SimCLR learns features from both left and right sides, whereas SL mainly learns label-relevant features from the left side of frequent data and ignore label-irrelevant features on the right side. In Figure 4, we provide results the high-resolution images of the 10 CIFAR classes to make the results easier to interpret. Here we further visualize the results on original CIFAR-10 images.

model on the loss without reweighting $\widehat{L}(\phi)$ by $\phi = \phi - \eta \nabla_\phi \widehat{L}(\phi + \epsilon(\phi))$. A detailed algorithm can be viewed in Algorithm 1.

---

**Algorithm 1** Reweighted Sharpness-Aware Minimization (rwSAM)

---

1: **Input:** the pre-training dataset $\widehat{\mathcal{D}}_s$.
2: **Output:** learned representations $\phi$.
3: **Stage 1:** compute the weight $w$.
4: **for** $i = 0$ **to** `MaxIter` **do**
5:     Randomly sample a batch of examples $\{x_i\}_{i=1}^b$ from $\widehat{\mathcal{D}}_s$.
6:     Update the representations $\phi$ on $\{x_i\}_{i=1}^b$ to minimize the loss.
$$\phi \leftarrow \phi - \eta \nabla_\phi \widehat{L}(\phi).$$
7: **end for**
8: Generate the weight with kernel density estimation:
$$w_i = \big(\frac{1}{n} \sum_{j=1}^n K(f_\phi(x_i) - f_\phi(x_j), h)\big)^{-\alpha}.$$
9: **Stage 1:** reweighted SAM.
10: **for** $i = 0$ **to** `MaxIter` **do**
11:     Randomly sample a batch of examples $\{x_i\}_{i=1}^b$ from $\widehat{\mathcal{D}}_s$.
12:     Calculate $\epsilon(\phi)$ based on the reweighted loss $\widehat{L}_w(\phi)$.
$$\epsilon_\phi = \rho \text{sgn}(\nabla_\phi \widehat{L}_w(\phi)) \left| \nabla_\phi \widehat{L}_w(\phi) \right|^{q-1} / \left( \|\nabla_\phi \widehat{L}_w(\phi)\|_q^q \right)^{1/p}$$
13:     Update the representations $\phi$ on $\{x_i\}_{i=1}^b$ to minimize the loss and penalize the sharpness,
$$\phi \leftarrow \phi - \eta \nabla_\phi \widehat{L}(\phi + \epsilon(\phi)).$$
14: **end for**

---

We select the hyperparameters $\rho$ and $\alpha$ with cross validation. On ImageNet-LT and iNaturalist, $\rho = 2$ and $\alpha = 0.5$. On CIFAR-10-LT, $\rho = 5$ and $\alpha = 1.2$.

## C.2 ADDITIONAL RESULTS

We further introduce another evaluation protocol of the representations learned on imbalanced ImageNet: following the protocol of Kang et al. (2020); Yang & Xu (2020), we fine-tune the representations on *imbalanced* ImageNet dataset with supervision, and then re-train the linear classifier with *class-aware resampling*, to compare with supervised imbalanced recognition methods. In MoCo V2 pre-training, we use the standard data augmen-

Table 3: ImageNet-LT with Supervision.

| Method | Backbone | Acc. |
|---|---|---|
| Supervised | ResNet-50 | 49.3 |
| CRT (Kang et al., 2020) | ResNet-50 | 52.0 |
| LADE (Hong et al., 2021) | ResNeXt-50 | 53.0 |
| RIDE (Wang et al., 2021) | ResNet-50 | 54.9 |
| RIDE (Wang et al., 2021) | ResNeXt-50 | 56.4 |
| MoCo V2 | ResNet-50 | 55.0 |
| MoCo V2+rwSAM | ResNet-50 | 55.5 |

tation following He et al. (2020). In fine-tuning, we use RandAugment (Cubuk et al., 2020). For this evaluation, we further compare with CRT (Kang et al., 2020), LADE (Hong et al., 2021), and RIDE (Wang et al., 2021), which are strong methods tailored to supervised imbalanced recognition. Results are provided in Table 3. Supervised here refers to training the feature extractor and linear classifier with supervision on the imbalanced dataset directly. CRT first trains the feature extractor with supervision, and then *re-trains the classifier with class-aware resampled loss*. Note that CRT is performing better than Supervised, indicating that the composition of the head and features learned from supervised learning is more sensitive to imbalanced dataset than the quality of feature extractor itself.

Even with a simple pre-training and fine-tuning pipeline, MoCo V2 representations can be comparable with much more complicated state-of-the-arts tailored to supervised imbalanced recognition, further corroborating the power of SSL under class imbalance. With rwSAM, we can further improve the result of MoCo V2.

## D  ADDITIONAL RELATED WORK

### D.1  SUPERVISED LEARNING WITH DATASET IMBALANCE

There exists a long line of works studying supervised imbalanced classification (He & Garcia, 2009; Krawczyk, 2016). Early works on ensemble learning adjusted the boosting and bagging algorithms with resampling in the imbalanced setting (Guo & Viktor, 2004; Wang & Yao, 2009). Classical methods include resampling and reweighting. Hart (1968); Kubat et al. (1997); Chawla et al. (2002); He et al. (2008); Ando & Huang (2017); Buda et al. (2018); Hu et al. (2020) proposed to re-sample the data to make the frequent and rare classes appear with equal frequency in training. Re-weighting assigns different weights for head and tail classes and eases the optimization difficulty under class imbalance (Cui et al., 2019; Tang et al., 2020; Wang et al., 2017b; Huang et al., 2019). Byrd & Lipton (2019) empirically studied the effect of importance weighting and found out that importance weighting does not change the solution without regularization. Xu et al. (2021) justified this finding with theoretical analysis based on the implicit bias of gradient descend on separable data.

Cao et al. (2019) initiated the idea of using re-weighted regularization and proposed the principle of regularizing rare classes more heavily. Re-weighted regularizaton is shown to be typically more effective than re-weighting or re-sampling the losses. Cao et al. (2021) proposed to regularize the local curvature of loss on imbalanced and noisy datasets.

Works in the modern deep learning era also designed specific losses or training pipelines for imbalanced recognition (Tang et al., 2020; Hong et al., 2021; Wang et al., 2021; Zhang et al., 2021). Lin et al. (2017) proposed to focus on hard examples to prevents easy examples from overwhelming the models during training. Meta-learning approaches meta-learned the weight or the ensemble (Wang et al., 2017b; Ren et al., 2018; Shu et al., 2019; Lee et al., 2020a). Liu et al. (2019); Jamal et al. (2020); Liu et al. (2020) improved the performance on the rare examples by explicitly encourages transfer learning. Re-calibration methods adjust the logits of the outputs with re-weighting (Tian et al., 2020; Menon et al., 2021).

Several works also studied the supervised representations under dataset imbalance. Kang et al. (2020); Wang et al. (2020) found out that the representations of supervised learning perform better than the classifier itself with class imbalance. Yang & Xu (2020) studied the effect of self-training and self-supervised pre-training on supervised imbalanced recognition classifiers. In contrast, the focus of our paper is the effect of class imbalance on *self-supervised representations*.

### D.2  SELF-SUPERVISED LEARNING

Earlier works on self-supervised learning learned visual representations by context prediction (Doersch et al., 2015; Wang et al., 2017a), solving puzzles (Noroozi & Favaro, 2016), and rotation prediction (Gidaris et al., 2018). Recent works on self-supervised learning successfully learn representations that approach the supervised baseline on ImageNet and various downstream tasks, and closed the gap with supervised pre-training. Contrastive learning methods attract positive pairs and drive apart negative pairs (He et al., 2020; Chen et al., 2020). Siamese networks predict the output of the other branch, and use stop-gradient to avoid collapsing (Grill et al., 2020; Chen & He, 2021). Clustering methods learn representations by performing clustering on the representations and

improve the representations with cluster index (Caron et al., 2020). Cole et al. (2021) investigated the effect of data quantity and task granularity on self-supervised representations. Goyal et al. (2021) studied self-supervised methods on large scale datasets in the wild, but they do not consider dataset imbalance explicitly. Kotar et al. (2021) studied whether dataset imbalance can have a significant impact on contrastive learning representations. Madaan et al. (2022) found out that self-supervised representations are better at continual learning than supervised representations. Several works have also theoretically studied the success of self-supervised learning (Arora et al., 2019; HaoChen et al., 2021; Wei et al., 2021; Lee et al., 2020b; Tian et al., 2021; Tosh et al., 2020; 2021). Our analysis in Section 3.1 is partially inspired by the work HaoChen et al. (2020).

# E  PROOF OF THEOREM 3.1

We notate data from the first class as $x_i^{(1)} = e_1 - q_i^{(1)}\tau e_2 + \rho\xi_i^{(1)}$ where $i \in [n_1]$ and $q_i^{(1)} \in \{0,1\}$. Similarly, we notate data from the second class as $x_i^{(2)} = -e_1 - q_i^{(2)}\tau e_2 + \rho\xi_i^{(2)}$ where $i \in [n_2]$ and $q_i^{(1)} \in \{0,1\}$. We notate data from the third class as $x_i^{(3)} = e_2 + \rho\xi_i^{(3)}$ where $i \in [n_3]$. Notice that all $\xi_i^{(k)}$ are independently sampled from $\mathcal{N}(0, I)$.

We first introduce the following lemma, which gives some high probability properties of independent Gaussian random variables.

**Lemma E.1.** *Let $\xi_i \sim \mathcal{N}(0, I)$ for $i \in [n]$. Then, for any $n \leq \mathrm{poly}(d)$, with probability at least $1 - e^{-d^{\frac{1}{10}}}$ and large enough $d$, we have:*

- $|\langle \xi_i, e_1 \rangle| \leq d^{\frac{1}{10}}$, $|\langle \xi_i, e_2 \rangle| \leq d^{\frac{1}{10}}$ *and* $|\|\xi_i\|_2^2 - d| \leq 4d^{\frac{3}{4}}$ *for all $i \in [n]$.*

- $|\langle \xi_i, \xi_j \rangle| \leq 3d^{\frac{3}{5}}$ *for all $i \neq j$.*

*Proof of Lemma E.1.* Let $\xi, \xi' \sim \mathcal{N}(0, I)$ be two independent random variables. By the tail bound of normal distribution, we have

$$\Pr\left(|\langle \xi, e_1 \rangle| \geq d^{\frac{1}{10}}\right) \leq d^{-\frac{1}{10}} \cdot e^{-\frac{d^{\frac{1}{5}}}{2}}. \tag{4}$$

By the tail bound of $\chi_d^2$ distribution, we have

$$\Pr\left(|\|\xi\|_2^2 - d| \geq 4d^{\frac{3}{4}}\right) \leq 2e^{-\sqrt{d}}. \tag{5}$$

Since the directions of $\xi$ and $\xi'$ are independent, we can bound their correlation with the norm of $\xi$ times the projection of $\xi'$ onto $\xi$:

$$\Pr\left(|\langle \xi, \xi' \rangle| \geq 3d^{\frac{3}{5}}\right) \leq \Pr\left(\|\xi\|_2 \geq \sqrt{d} + 2d^{\frac{3}{8}}\right) + \Pr\left(|\langle \xi', \frac{\xi}{\|\xi\|}\rangle| \geq d^{\frac{1}{10}}\right) \leq \frac{e^{-\frac{d^{\frac{1}{5}}}{2}}}{d^{\frac{1}{10}}} + 2e^{-\sqrt{d}}. \tag{6}$$

Since every $\xi_i$ and $\xi_j$ are independent when $i \neq j$, by the union bound, we know that with probability at least $1 - (n^2 + 2n)(\frac{e^{-\frac{d^{\frac{1}{5}}}{2}}}{d^{\frac{1}{10}}} + 2e^{-\sqrt{d}})$, we have $|\langle \xi_i, e_1 \rangle| \leq d^{\frac{1}{10}}$, $|\langle \xi_i, e_2 \rangle| \leq d^{\frac{1}{10}}$ and $|\|\xi_i\|_2^2 - d| \leq 4d^{\frac{3}{4}}$ for all $i \in [n]$, and also $|\langle \xi_i, \xi_j \rangle| \leq 3d^{\frac{3}{5}}$ for all $i \neq j$. Since the error probability is exponential in $d$, for large enough $d$, the error probability is smaller than $e^{-d^{\frac{1}{10}}}$, which finishes the proof. $\square$

Using the above lemma, we can prove the following lemma which constructs a linear classifier of the empirical dataset with relatively large margin and small norm.

**Lemma E.2.** *In the setting of Theorem 3.1, let $w_1^* = e_1$, $w_2^* = -e_1$, $w_3^* = \frac{1}{\rho d}\sum_{i=1}^{n_3} \xi_i^{(3)}$. Apply Lemma E.1 to the set of all $\xi_i^{(k)}$ where $k \in [3]$ and $i \in [n_k]$. When the high probability outcome of Lemma E.1 happens, the margin of classifier $\{w_1^*, w_2^*, w_3^*\}$ is at least $1 - O(d^{-\frac{1}{10}})$. Furthermore, we have $\|w_3^*\|_2^2 \leq O(d^{-\frac{1}{5}})$.*

*Proof of Lemma E.2.* When the high probability outcome of Lemma E.1 happens, we give a lower bound on the margin for all data in the dataset. For data $x = x_i^{(1)}$ in class 1, we have

$$w_1^{*\top} x = 1 + \langle \xi_i^{(1)}, e_1 \rangle\rho \geq 1 - \rho d^{\frac{1}{10}}, \tag{7}$$

$$w_2^{*\top} x = -1 + \langle \xi_i^{(1)}, e_1 \rangle\rho \leq -1 + \rho d^{\frac{1}{10}}, \tag{8}$$

$$w_3^{*\top} x = \frac{1}{\rho d}\left(e_1 - q_i^{(1)}\tau e_2 + \rho\xi_i^{(1)}\right)^{\top}\left(\sum_{j=1}^{n_3} \xi_j^{(3)}\right) \leq \frac{n_3(\tau+1)}{\rho d}d^{\frac{1}{10}} + \frac{3n_3}{d}d^{\frac{3}{5}}. \tag{9}$$

So the margin on data $(x_i^{(1)}, 1)$ is

$$w_1^{*\top} x - w_3^{*\top} x \geq 1 - \rho d^{\frac{1}{10}} - \frac{n_3(\tau+1)}{\rho d} d^{\frac{1}{10}} - \frac{3n_3}{d} d^{\frac{3}{5}} \geq 1 - O(d^{-\frac{1}{10}}). \quad (10)$$

Similarly, for data $x_i^{(2)}$ in class 2, the margin is at least $1 - O(d^{-\frac{1}{10}})$.

For data $x = x_i^{(3)}$ in class 3, we have

$$w_3^{*\top} x = \frac{1}{\rho d} \left( \sum_{j=1}^{n_3} \xi_j^{(3)} \right)^\top \left( e_2 + \rho \xi_i^{(3)} \right) \geq \frac{1}{d} \|\xi_i^{(3)}\|_2^2 - \frac{3n_3}{d} d^{\frac{3}{5}} - \frac{n_3 d^{\frac{1}{10}}}{\rho d} \geq 1 - O(d^{-\frac{1}{5}}). \quad (11)$$

On the other hand,

$$w_1^{*\top} x = \langle \rho \xi_i^{(3)}, e_1 \rangle \leq \rho d^{\frac{1}{10}}, \quad (12)$$

$$w_2^{*\top} x = \langle \rho \xi_i^{(3)}, -e_1 \rangle \leq \rho d^{\frac{1}{10}}. \quad (13)$$

So the margin is

$$w_3^{*\top} x - \max\{w_1^{*\top} x, w_2^{*\top} x\} \geq w_3^{*\top} x - \rho d^{\frac{1}{10}} \geq 1 - O(d^{-\frac{1}{10}}). \quad (14)$$

Finally, noticing that $\|w_3^*\|_2 \leq \frac{2n_3\sqrt{d}}{\rho d} \leq 2d^{-\frac{1}{10}}$ finishes the proof. $\qquad \square$

We also introduce the following helper lemma:

**Lemma E.3.** *Let $W \in \mathbb{R}^{3 \times d}$ be an arbitrary matrix, $m \geq 3$. Then, we have*

$$\|W\|_F^2 = \frac{1}{2} \min_{W_2 W_1 = W} \left( \|W_2^\top W_2\|_F^2 + \|W_1 W_1^\top\|_F^2 \right), \quad (15)$$

*where $W_1 \in \mathbb{R}^{m \times d}$ and $W_2 \in \mathbb{R}^{3 \times m}$. Furthermore, the minimum is achieved when $W_1 W_1^\top = W_2^\top W_2$.*

*Proof.* On one hand, we have

$$\|W\|_F^2 = Tr(WW^\top) \quad (16)$$

$$= \min_{W_2 W_1 = W} Tr(W_2 W_1 W_1^\top W_2^\top) \quad (17)$$

$$= \min_{W_2 W_1 = W} Tr(W_1 W_1^\top W_2^\top W_2) \quad (18)$$

$$\leq \frac{1}{2} \min_{W_2 W_1 = W} (\|W_1 W_1^\top\|_F^2 + \|W_2^\top W_2\|_F^2), \quad (19)$$

where the inequality becomes equality if and only if $W_1 W_1^\top = W_2^\top W_2$.

On the other hand, let $W = U\Sigma V$ be the SVD decomposition of $W$, where $\Sigma \in \mathbb{R}^{3 \times d}$ is a diagonal matrix with $\sigma_1, \sigma_2, \sigma_3$ on its diagonal. For integers $p, q \geq 3$, we use $\Sigma_{p \times q}^{\frac{1}{2}}$ to denote the $p \times q$ matrix with $\sqrt{\sigma_1}, \sqrt{\sigma_2}, \sqrt{\sigma_3}$ at its first 3 diagonal positions and 0 otherwise. If we set $W_1 = \Sigma_{m \times d}^{\frac{1}{2}} V$ and $W_2 = U\Sigma_{3 \times m}^{\frac{1}{2}}$, then it can be verified that $W = W_2 W_1$ and $\|W\|_F^2 = \frac{1}{2}(\|W_1 W_1^\top\|_F^2 + \|W_2^\top W_2\|_F^2)$. Therefore, the equality holds in Equation 19, which finishes the proof. $\qquad \square$

Now we are ready to prove the supervised learning part of Theorem 3.1:

*Proof of Theorem 3.1 (supervised learning part).* Let $\{\hat{w}_1, \hat{w}_2, \hat{w}_3\}$ be three vectors in $\mathbb{R}^d$ that minimize $\|w_1\|_2^2 + \|w_2\|_2^2 + \|w_3\|_2^2$ subject to the margin constraint $w_y^\top x \geq w_{y'}^\top x + 1$ for all empirical data $(x, y)$ and $y' \neq y$. To prove the supervised learning part of Theorem 3.1, we will first prove that $\langle \hat{w}_1, e_1 \rangle^2 + \langle \hat{w}_2, e_1 \rangle^2 + \langle \hat{w}_3, e_1 \rangle^2 \leq O(d^{-\frac{1}{10}})$ with high probability, and then use this result to prove the correlation between $e_2$ and $W_{\text{SL}}$.

We frist apply Lemma E.1 to the set of all $\xi_i^{(k)}$ where $k \in [3]$ and $i \in [n_k]$. We consider the situation when the high probability outcome of Lemma E.1 holds (which happens with probability at least $1 - e^{-d^{\frac{1}{10}}}$). By Lemma E.2, the constructed classifier $\{w_1^*, w_2^*, w_3^*\}$ has margin $\alpha \geq 1 - O(d^{-\frac{1}{10}})$ in this case. As a result, $\{\frac{1}{\alpha}w_1^*, \frac{1}{\alpha}w_2^*, \frac{1}{\alpha}w_3^*\}$ is a classifier with margin 1 and norm bounded by

$$\|\frac{1}{\alpha}w_1^*\|_2^2 + \|\frac{1}{\alpha}w_2^*\|_2^2 + \|\frac{1}{\alpha}w_3^*\|_2^2 = \frac{2 + \|w_3^*\|_2^2}{\alpha^2} \leq 2 + O(d^{-\frac{1}{10}}). \tag{20}$$

Let $\{\hat{w}_1, \hat{w}_2, \hat{w}_3\}$ be min-norm linear classifier of the empirical dataset. Since its norm cannot be larger than the constructed one, we have $\|\hat{w}_1\|_2^2 + \|\hat{w}_2\|_2^2 + \|\hat{w}_3\|_2^2 \leq 2 + O(d^{-\frac{1}{10}})$. By standard concentration inequality, when $n_1 \geq \text{poly}(d)$, with probability at least $1 - e^{d^{-\frac{1}{10}}}$, we have

$$\left| \mathbb{E}_{i \in [n_1], q_i^{(1)}=0}[x_i^{(1)}] - e_1 \right| \leq d^{-\frac{1}{10}}, \tag{21}$$

where the expectation is over all the data from class 1 that satisfies $q_i^{(1)} = 0$. By the definition of $\{\hat{w}_1, \hat{w}_2, \hat{w}_3\}$ we know $(\hat{w}_1 - \hat{w}_3)^\top x_i^{(1)} \geq 1$ for all $i \in [n_1]$, hence averaging over all the class 1 data with $q_i^{(1)} = 0$ and using the above inequality gives us

$$(\hat{w}_1 - \hat{w}_3)^\top e_1 \geq 1 - \|\hat{w}_1 - \hat{w}_3\|_2 \cdot d^{-\frac{1}{10}} \geq 1 - O(d^{-\frac{1}{10}}). \tag{22}$$

A similar analysis for class 2 data gives us

$$(\hat{w}_2 - \hat{w}_3)^\top (-e_1) \geq 1 - O(d^{-\frac{1}{10}}). \tag{23}$$

Now we prove that $\hat{w}_1, \hat{w}_2, \hat{w}_3$ all have small correlation with $e_2$. Without loss of generality, we assume $\hat{w}_3^\top e_1 \triangleq t \geq 0$. If $t \geq \frac{1}{2}$, we have

$$\langle \hat{w}_1, e_1 \rangle^2 + \langle \hat{w}_2, e_1 \rangle^2 + \langle \hat{w}_3, e_1 \rangle^2 \geq \left( t + 1 - O(d^{-\frac{1}{10}}) \right)^2 > 2.25 - O(d^{-\frac{1}{10}}), \tag{24}$$

which contradicts with $\|\hat{w}_1\|_2^2 + \|\hat{w}_2\|_2^2 + \|\hat{w}_3\|_2^2 \leq 2 + O(d^{-\frac{1}{10}})$. Therefore, there must be $t \leq \frac{1}{2}$, hence

$$\langle \hat{w}_1, e_1 \rangle^2 + \langle \hat{w}_2, e_1 \rangle^2 + \langle \hat{w}_3, e_1 \rangle^2 \tag{25}$$

$$\geq \left( 1 + t - O(d^{-\frac{1}{10}}) \right)^2 + \left( 1 - t - O(d^{-\frac{1}{10}}) \right)^2 + t^2 \tag{26}$$

$$\geq 2 + 3t^2 - O(d^{-\frac{1}{10}}) \tag{27}$$

$$\geq 2 - O(d^{-\frac{1}{10}}). \tag{28}$$

As a result,

$$\langle \hat{w}_1, e_2 \rangle^2 + \langle \hat{w}_2, e_2 \rangle^2 + \langle \hat{w}_3, e_2 \rangle^2 \tag{29}$$

$$\leq \|\hat{w}_1\|_2^2 + \|\hat{w}_2\|_2^2 + \|\hat{w}_3\|_2^2 - \langle \hat{w}_1, e_1 \rangle^2 - \langle \hat{w}_2, e_1 \rangle^2 - \langle \hat{w}_3, e_1 \rangle^2 \tag{30}$$

$$\leq \left( 2 + O(d^{-\frac{1}{10}}) \right) - \left( 2 - O(d^{-\frac{1}{10}}) \right) \tag{31}$$

$$\leq O(d^{-\frac{1}{10}}). \tag{32}$$

Now we turn to the analysis of $W_{\text{SL}}$. Recall that we learn two matrices $W_1 \in \mathbb{R}^{m \times d}$ and $W_2 \in \mathbb{R}^{3 \times m}$ that minimize $\|W_1^\top W_1\|_F^2 + \|W_2^\top W_2\|_F^2$ subject to the margin constraint $(W_2 W_1 x)_y \geq (W_2 W_1 x)_{y'} + 1$, and the supervised representation is $W_{\text{SL}} = W_1$. According to Lemma E.3, we know that the solution $W_1$ and $W_2$ satisfy $W_2 W_1 = [\hat{w}_1, \hat{w}_2, \hat{w}_3]^\top$ and $W_2^\top W_2 = W_1 W_1^\top$. Let $W_2^\top W_2 = W_1 W_1^\top = U^\top \Sigma U$ be the SVD decomposition, where $\Sigma \in \mathbb{R}^{m \times m}$ is a non-negative diagonal matrix and $U$ is a unitary matrix. Since $W_2$ has rank at most 3, there are at most 3 entries in $\Sigma$ that are non-zero. Without loss of generality, we assume that all the non-zero entries of $\Sigma$ are in the first 3 rows.

Let $\Sigma_{m \times d}$ and $\Sigma_{3 \times m}$ be the matrices by reshaping $\Sigma$ (deleting or padding all-0 rows/columns) to the corresponding dimensions. We can write $W_1$ as $W_1 = U^\top \Sigma_{m \times d}^{\frac{1}{2}} V_1$ for some unitary matrix

$V_1 \in \mathbb{R}^{d \times d}$, where $\Sigma_{m \times d}^{\frac{1}{2}}$ is the element-wise square root of $\Sigma_{m \times d}$. Similarly, $W_2 = V_2 \Sigma_{3 \times m}^{\frac{1}{2}} U$ for some unitary matrix $V_2 \in \mathbb{R}^{3 \times 3}$. Taking the product gives $W_2 W_1 = V_2 \Sigma_{3 \times d} V_1$.

Let $W_{SL} = W_1 = [w_1, w_2, \cdots, w_m]^\top$. Now we finishe the proof with

$$\sum_{i=1}^m \langle w_i, e_2 \rangle^2 = \|W_1 e_2\|_2^2 = \|U^\top \Sigma_{m \times d}^{\frac{1}{2}} V_1 e_2\|_2^2 \le \|V_1 e_2\|_2^2 \cdot \|\Sigma_{d \times d} V_1 e_2\|_2^2 = \|V_2 \Sigma_{3 \times d} V_1 e_2\|_2^2 \tag{33}$$

$$= \|W_2 W_1 e_2\|_2^2 = \langle \hat{w}_1, e_2 \rangle^2 + \langle \hat{w}_2, e_2 \rangle^2 + \langle \hat{w}_3, e_2 \rangle^2 \le O(d^{-\frac{1}{10}}). \tag{34}$$

$\square$

To prove the self-supervised learning part of Theorem 3.1, we first introduce the following lemma which gives some helpful properties of the empirical data matrix.

**Lemma E.4.** *In the setting of Theorem 3.1, let $M \triangleq \mathbb{E}_x[xx^\top]$ where the expectation is over empirical data. Then, when $n_1, n_2 \ge \text{poly}(d)$, with probability at least $1 - e^{-d^{\frac{1}{10}}}$, we have: (1) $e_2^\top M e_2 \ge \Omega(d^{\frac{2}{5}})$, and (2) $u^\top M u \le O(1)$ for all $u \in \mathbb{R}^d$ such that $u^\top e_2 = 0$ and $\|u\|_2 = 1$.*

*Proof of Lemma E.4.* Let $n = n_1 + n_2 + n_3$. We abuse notation and let $\xi_i$ ($i \in [n]$) be the set of all $\xi_i^{(k)}$ that appears in the empirical data. Let matrix $M' = \frac{1}{n} \sum_{i=1}^n \xi_i \xi_i^\top$. By standard concentration inequalities and union bound, for $n \ge \text{poly}(d)$, with probability at least $1 - \frac{1}{2} e^{-d^{\frac{1}{10}}}$, we have that $|M'_{i,j}| \le \frac{1}{d}$ for all $i \ne j$ and $|M'_{i,i} - 1| \le \frac{1}{d}$ for all $i \in [d]$. In this case, for any vector $u \in \mathbb{R}^d$ such that $\|u\|_2 = 1$ and $u^\top e_2 = 0$, we have

$$u^\top M u \le 2\|u\|_2^2 + 2u^\top(\rho^2 M')u \le 2 + 2\rho^2 + 2\rho^2 \|M' - I\|_F \le O(1). \tag{35}$$

On the other hand, by the definition of data distirbution and standard concentration inequalities, for $n \ge \text{poly}(d)$, with probability at least $1 - \frac{1}{2} e^{-d^{\frac{1}{10}}}$ we have that: at least $\frac{1}{3}$ of all data either is class 1 with $q_i^{(1)} = 1$ or class 2 with $q_i^{(2)} = 1$, and $\|\frac{1}{n} \sum_{i=1}^n \xi_i\|_2 \le O(\frac{1}{d})$. In this case,

$$e_2^\top M e_2 = \mathbb{E}_x[(e_2^\top x)^2] \ge (\mathbb{E}_x[e_2^\top x])^2 \ge \left( \frac{1}{3}\tau - e_2^\top \left( \frac{1}{n} \sum_{i=1}^n \xi_i \right) \right)^2 \ge \Omega(\tau^2) = \Omega(d^{\frac{2}{5}}). \tag{36}$$

$\square$

Using the above lemma, we can prove the self-supervised learning part of Theorem 3.1.

*Proof of Theorem 3.1(self-supervised learning part).* Let $M = \mathbb{E}_x[xx^\top]$ be the empirical data matrix, where the expectation is over the dataset. Notice that self-supervised learning objective has the same minimizer as the matrix factorization objective $\|M - W^\top W\|_F^2$, by Eckart–Young–Mirsky theorem we know that the span of $\tilde{w}_1, \tilde{w}_2, \cdots, \tilde{w}_m$ is exactly the span of the top $m$ eigenvectors of matrix $M$. Let $M = \sum_{i=1}^d \lambda_i v_i v_i^\top$ where $\lambda_i$ is the $i$-th largest eigenvalue of $M$ with the corresponding eigenvector $v_i$. We decompose $e_2$ in the eigenvector basis as $e_2 = \sum_{i=1}^d \zeta_i v_i$.

We first note that $\lambda_1 \ge \Omega(d^{\frac{2}{5}})$ and $\max_{i \ne 1} \lambda_i \le O(1)$. Indeed, we know that $\mathbb{E}[M] = \text{diag}(1 + d^{-\frac{2}{5}}, d^{\frac{2}{5}} + d^{-\frac{2}{5}}, d^{-\frac{2}{5}}, \cdots, d^{-\frac{2}{5}})$. By standard matrix concentration bounds (e.g. Theorem 4.6.1 of Vershynin (2018)), we know that with probability at least $1 - e^{-d^{\frac{1}{10}}}$, $\|M - \mathbb{E}[M]\| \le O(d^{-\frac{2}{5}})$. By Weyl's inequality we know that $\max_i |\lambda_i(T) - \lambda_i(S)| \le \|S - T\|$, so $\lambda_1 \ge \Omega(d^{\frac{2}{5}})$ and $\max_{i \ne 1} \lambda_i \le O(1)$.

By Lemma E.4, we know that with probability at least $1 - e^{-d^{\frac{1}{10}}}$, we have $e_2^\top M e_2 \ge \Omega(d^{\frac{2}{5}})$ and $\frac{u^\top M u}{\|u\|_2^2} \le O(1)$ for all $u$ orthogonal to $e_2$. To prove the result regarding self-supervised learning, we only need to prove that $\zeta_1^2 \ge 1 - O(d^{-\frac{1}{5}})$ in this case.

We first show that $\zeta_1^2 \geq \frac{1}{2}$. For contradiction, first assume $\zeta_1^2 \leq \frac{1}{2}$. Define vector

$$u = \sqrt{1 - \zeta_1^2} v_1 - \frac{\zeta_1 \sum_{i=2}^{d} \zeta_i v_i}{\sqrt{1 - \zeta_1^2}}. \tag{37}$$

which satisfies $u^\top e_2 = 0$ and $\|u\|_2^2 = 1$. Notice that

$$\frac{u^\top M u}{\|u\|_2^2} = (1 - \zeta_1^2)\lambda_1 - \frac{\zeta_1^2 \sum_{i=2}^{d} \zeta_i^2 \lambda_i}{1 - \zeta_1^2} \tag{38}$$

$$\geq \frac{\lambda_1}{2} - \max_{i \neq 1} \lambda_i \tag{39}$$

$$\geq \Omega(d^{\frac{2}{5}}), \tag{40}$$

which contradicts to $\frac{u^\top M u}{\|u\|_2^2} \leq O(1)$. Therefore, we have $\zeta_1^2 \geq \frac{1}{2}$.

To prove that $\zeta_1^2$ is close to 1, we let scalar $t = \frac{1}{\zeta_1^2} - 1$ and define vector

$$u = -t\zeta_1 v_1 + \sum_{i=2}^{d} \zeta_i v_i, \tag{41}$$

which satisfies $u^\top e_2 = 0$. Since $\zeta_1^2 \geq \frac{1}{2}$, we have $t \leq 1$ and $\|u\|_2^2 \leq 1$. As a result, we have

$$\frac{u^\top M u}{\|u\|_2^2} \geq t^2 \lambda_1 \zeta_1^2 + \sum_{i=2}^{d} \lambda_i \zeta_i^2 \geq t^2 e_2^\top M e_2 \geq \Omega(t^2 d^{\frac{2}{5}}). \tag{42}$$

On the other hand, we know that $\frac{u^\top M u}{\|u\|_2^2} \leq O(1)$. Comparing these two bounds gives us $t^2 \leq O(d^{-\frac{1}{5}})$, which means $\zeta_1^2 \geq 1 - O(d^{-\frac{1}{5}})$. $\qquad\square$

