# OpenReview forum: "Self-supervised Learning is More Robust to Dataset Imbalance"
_ICLR.cc/2022/Conference — ICLR 2022 Spotlight_

### Official Review · Reviewer_qJjn · 2021-10-30

**Correctness:** 4
**Technical Novelty And Significance:** 3
**Empirical Novelty And Significance:** 4
**Recommendation:** 8
**Confidence:** 4

**Main Review:**

I enjoyed the paper. It reads well and has a robust set of experiments to investigate the effects of class imbalance on representation learning. Some suggestions and questions on how to improve the paper:

- The effect of class difficulty: while random sampling for rare classes is a good strategy to take, I wonder how the class difficulty may play a role. This may become important as usually rare classes are also harder classes (even for annotators). So, an experimental setup to test this would be to use the class performance of a model on a balanced dataset to deliberately more frequently choose challenging classes as rare.
- reSAM computational complexity: because of pairwise kernel within the regularisation term, I think this methods can't scale to very large datasets. So, what are your ideas on how to improve this? I thought of in-batch sampling, but that might be ineffective as it's possible samples in a batch are already far away from each other.

Minor comments:
- Abstract - first sentence: While the paper focuses on vision, SSL is not a vision specific. For example, NLP domain has been enjoying SSL since ELMo, BERT, ...
- NLP applications: following from above, would be interesting to study the effect of dataset balance on NLP applications where SSL is a dominant pre-training approach.

**Summary Of The Paper:**

This paper studies and compares the performance of self-supervised representation learning methods against supervised representation learning when there are class imbalance. It shows empirically that self-supervised methods are more robust to class imbalance, i.e. the performance gap between models trained on balanced and imbalanced datasets is smaller. The paper also investigates the quality of the features learnt by SSL vs SL, and shows both theoretically in a limited setting and empirically on synthetic datasets that features learnt by SSL are diverse and capture characteristics that might be useful for rare classes. Finally, they provide a regularisation method for SSL to encourage model to learn better features for rare classes without having access to labels.

**Summary Of The Review:**

I think this is an important paper with experiments and limited theory to show and explain why SSL methods are more robust to class imbalance problem which also occur frequently in practice.

---

> ### Author Response · Authors · 2021-11-20
> **Response to Reviewer qJjn**
>
> Thanks for providing constructive feedback, and for noting that our paper “has a robust set of experiments”. We address the comments below.
>
> **Q1**: The effect of class difficulty.
> **A1**: Following the suggestion, we resample the ImageNet to make the harder classes rarer. Concretely, we rank the classes based on their validation accuracy in the balanced case, and use the classes with low validation frequency as the rare classes. However, results indicate that this does not result in significant performance drop. In the table below, “ImageNet, imbalanced, hard” refers to the above protocol. “ImageNet, imbalanced, random” refers to the random selection of rare classes in our paper. We hypothesize that 1) the difficulty of ImageNet classes do not vary much and 2) the harder classes do not matter much to the downstream tasks.
>
> | Datasets | OOD performance of MoCo v2 |
> | :--- | :---: |
> | ImageNet, balanced | 80.3 |
> | ImageNet, imbalanced, random | 80.2 |
> | ImageNet, imbalanced, hard | 80.0 |
>
>
>
> **Q2**: The computational complexity of rwSAM.
> **A2**: In our implementation, we calculate the kernel matrix on the GPU with Pytorch. We use block matrix multiplication to lower the memory use. On a single 2080Ti, the calculation of the kernel density estimation can be done in less than 30 minutes on ImageNet-LT, which is negligible compared to training the networks. For even larger datasets, one can consider using Cholesky decomposition or Nyström method [1] to accelerate the computation of kernel matrices.
>
> **Q3**: Class imbalance datasets in NLP.
> **A3**: We admit that NLP is a very important SSL application. However, NLP pre-training datasets do not have an explicit notion of “classes” as in vision datasets, so it is hard to define “class imbalance”. We thought about mixing different domains of text (such as wikitext and openwebtext) with different ratios to simulate the imbalance, but the data quality of each domain can be very different, which may complicate the effect of imbalance itself. We will leave the NLP applications for future work.
>
>
> [1] Williams, Christopher, and Matthias Seeger. "Using the Nyström method to speed up kernel machines." Proceedings of the 14th annual conference on neural information processing systems.

---

> > ### Comment · Reviewer_qJjn · 2021-11-29
> > **re**
> >
> > Thanks for the added experiments and insights.
> >
> > On Q3, that's a good call and you may want to add that to your paper as NLP pre-training datasets are often not the same dataset that you do the final task fine-tuning.

---

### Official Review · Reviewer_8MoR · 2021-11-01

**Correctness:** 2
**Technical Novelty And Significance:** 2
**Empirical Novelty And Significance:** 2
**Recommendation:** 5
**Confidence:** 2

**Main Review:**

This paper focuses on class-imbalanced learning and shows that self-supervised pre-training yields representations that are more robust to dataset imbalance.

Strength:
1) This paper is well-written and the conclusion is clear.

Weakness:
1) The conclusions are mainly observed from empirical results. There lack of theoretical insights.
2) The generalization of the conclusions is not clear. Given an imbalanced dataset, how to decide which algorithm should I adopt? For example, on what conditions self-supervised learning is better than supervised learning, and on what conditions are not?
3) Some class-imbalanced learning methods are not compared. In section 4.1, the authors report the performance of the proposed rwSAM on imbalanced datasets. More related algorithms should be compared to show the effectiveness of the proposal.


**Summary Of The Paper:**

This paper focuses on class-imbalanced learning and shows that self-supervised pre-training yields representations that are more robust to dataset imbalance.


**Summary Of The Review:**

This paper focuses on class-imbalanced learning and shows that self-supervised pre-training yields representations that are more robust to dataset imbalance. The problem is interesting. But the results are mainly observed from empirical results, the generalization of the conclusions is not clear. The novelty and contribution of the paper are limited.

---

> ### Author Response · Authors · 2021-11-20
> **Response to Reviewer 8MoR**
>
> Thanks for taking the time reading our paper and providing detailed comments. We will address the concerns below.
>
> **Q1**: The conclusions are mainly observed from empirical results. There is a lack of theoretical insights.
> **A1**: In the original submission, we actually provided a theoretical analysis in $\underline{\text{Section 3.1}}$ to justify our empirical results. We prove that on specific distributions, supervised learning only learns features that can classify the frequent classes, while self-supervised learning can further learn the label-irrelevant but transferable features which help the rare classes. The theoretical result here is consistent with the empirical analysis in $\underline{\text{Section 3.2}}$.
>
> **Q2**: The generalization of the conclusions is not clear. Given an imbalanced dataset, how to decide which algorithm should I adopt? For example, on what conditions self-supervised learning is better than supervised learning, and on what conditions are not?
> **A2**: In $\underline{\text{Section 2.3}}$, we discovered that, **when transferred to the target datasets**, self-supervised learning consistently outperforms supervised learning under pre-training dataset imbalance. This clearly indicates that when we have a large scale imbalanced dataset for pre-training and care about the **downstream performance**, we should use self-supervised learning.
>
>
> **Q3**: Some class-imbalanced learning methods are not compared.
> **A3**: Supervised class-imbalanced learning methods rely on access to labels to compute the re-weighting or adjust the training objective. In our SSL setting, we have no access to labels, which makes the supervised class-imbalanced learning methods **not applicable**. Nonetheless, in the original submission, we provided the comparison of our method with state-of-the-art supervised class-imbalanced learning methods in the supervised setting on ImageNet-LT in $\underline{\text{Section C.2}}$ of the appendix. We also put the results here. Even with a simple pre-training and fine-tuning pipeline, MoCo v2 representations can be comparable with much more complicated state-of-the-arts tailored to supervised imbalanced recognition, further corroborating the power of SSL under class imbalance. With rwSAM, we can further improve the result of MoCo v2.
>
> | Method | Backbone | Accuracy |
> | :--- | :---: | :---: |
> | Decoupling | ResNet-50 | 52.0 |
> | LADE | ResNeXt-50 | 53.0 |
> | RIDE | ResNet-50 | 54.9 |
> | MoCo v2 | ResNet-50 | 55.0 |
> | MoCo v2 + rwSAM | ResNet-50 | 55.5 |

---

### Official Review · Reviewer_Jyb6 · 2021-11-01

**Correctness:** 3
**Technical Novelty And Significance:** 3
**Empirical Novelty And Significance:** 3
**Recommendation:** 8
**Confidence:** 4

**Main Review:**

The paper is well written in general and its motivation is clear.

Weaknesses:

- The only task considered in the paper is image classification using the standard accuracy as the evaluation metric. Is the observed robustness also valid for other tasks? Although the paper presents a large range of experiments, the conclusions are less general than what is sold. The imbalanced datasets used in the paper are CIFAR-10-LT and ImageNet-LT, which are versions of CIFAR-10 and ImageNet that are originally balanced. Datasets that follow an intrinsic long-tail distribution (e.g., iNaturalist 2018) are not considered. Although the superior results of SSL in imbalanced settings are promising (e.g. the apparent robustness in Figure 2), it would be interesting to know if the conclusions in the paper generalize to datasets that are originally imbalanced (e.g., iNaturalist 2018).

- The improvement of the proposed regularization in Tables 1 and 2 does not seem statistically significant.

- The toy dataset setup in Section 3.1 is difficult to read.


Despite the weaknesses mentioned above, I think that the paper is worth publishing. It considers a problem that is relevant to the machine learning and computer vision communities. It shows that, in the OOD case or very imbalanced case, it is better to use SSL techniques (e.g. SimCLR or MoCo v2) since they do not overfit to the training samples and labels, and the imbalanced factor does not matter.
The paper gives a sensible explanation of why this is the case, with detailed ablation studies.
Although Table 2 uses diverse datasets such as CUB, Cars, Aircrafts and Pets, the improvement does not seem significant. It is then difficult to know whether or not the conclusions of the paper generalize to datasets other than CIFAR-10 and ImageNet.






minor typo in titles of Figure 1: "Imabalanced" => "Imbalanced"

**Summary Of The Paper:**

The paper studies the differences between representations pretrained with Self-Supervised Learning (SSL) and the standard Supervised Learning (SL) frameworks in the In-Domain (ID) and Out-Of-Domain (OOD) settings of the image classification task with an imbalanced dataset.

The paper presents different results that are relevant to the computer vision and machine learning community:
1) In the ID setting, representations pretrained with SL outperform those pretrained with SSL. However, in the OOD setting, representations pretrained with SSL outperform those pretrained with SL and are even robust to the imbalance factor.
2) An analysis in Section 3.1 on a toy dataset and semi-synthetic data (by cropping and fusing half images during pretraining) in Section 3.2 validate that SSL methods learn representations that do not overfit to labels and show improved accuracy at test time.
3) The paper also proposes a regularization framework (called rwSAM) that promotes flatter landscape for rare examples to improve performance at test time (see Table 1 and 2).

**Summary Of The Review:**

The paper considers the imbalanced dataset problem that is relevant to the machine learning and computer vision communities. The proposed analysis also makes sense and is interesting to the ML and CV communities.

---

> ### Author Response · Authors · 2021-11-20
> **Response to Reviewer Jyb6**
>
> Thanks for providing constructive comments! We will answer them as below.
>
> **Q1**: Is the observed robustness also valid for other tasks?
> **A1**: SSL is still more robust to dataset class imbalance when the downstream task is detection. We added the results of transferring the representations to the PascalVOC detection task. We still consider MoCo v2 and ImageNet-LT following the setting of $\underline{\text{Section 2.2}}$. During fine-tuning, we train the models on PascalVOC 07 and PascalVOC 12 training set and test on the PascalVOC 07 test set following the MoCo and SimCLR paper. As shown in the following table, the gap between imbalance and balanced pertaining with MoCo is much smaller than the gap with supervised learning across all numbers of examples.  We will add this result in the revision.
>
> | Number of examples | 116K | 58K | 14K |
> | :--- | :--: | :---: | :---: |
> | MoCo v2, balanced | 78.3 | 76.5 | 74.3 |
> | MoCo v2, imbalanced | 77.9 | 76.0 | 74.1 |
> | MoCo v2, gap | 0.2% | 0.7% | 0.3% |
> | Supervised, balanced | 74.8 | 71.4 | 61.0 |
> | Supervised, imbalanced | 74.0 | 69.2 | 60.5 |
> | Supervised, gap | 1.1% | 3.1% | 0.8% |
>
>
>
>
> **Q2**: Datasets that follow an intrinsic long-tail distribution (e.g., iNaturalist 2018) are not considered.
> **A2**: The focus of this paper is the robustness of SSL under class imbalance, which is measured by the relative performance gap of representations between balanced and imbalanced datasets with the same number of pre-training examples. To calculate this relative gap, one must have access to a balanced dataset and an imbalanced dataset with the same number of examples and the same class-conditional distribution. Therefore, we need a balanced dataset for reference and cannot conduct the same analysis on iNaturalist---we cannot convert iNaturalist 2018 into a balanced dataset with the same number of examples because the rare classes contain less than 5 examples. To indirectly address the question, we provide the results on iNaturalist 2018 in the table below and compare with results on the whole balanced ImageNet. iNaturalist 2018 (438K examples) is only 2.2% below the balanced ImageNet (1.2M examples) even with the imbalance, indicating that the performance is reasonably good.
>
>
> | Dataset | Number of examples | OOD performance of MoCo v2 % |
> | :--- | :---: | :---: |
> | ImageNet, imbalanced | 116K | 80.2 |
> | ImageNet, subsampled, balanced | 116K | 80.3 |
> | ImageNet, whole, balanced | 1.2M | 84.3 |
> | iNaturalist, imbalanced | 438K | 82.1 |
>
>
>
> **Q3**: The improvement of the proposed regularization in Tables 1 and 2 does not seem statistically significant.
> **A3**: We would like to clarify that the gap between balance and imbalanced datasets are already very small, as we argued in the earlier sections of the paper. Therefore, we should not expect an algorithm that addresses the imbalanced issue to be significantly better than even the balanced pretraining. Indeed, we note that “SimSiam, balanced” and “MoCo v2, balanced” refer to the models trained on the balanced dataset. Our method can close the small gap---we train on an imbalanced dataset but achieve performance comparable to or better than models trained on balanced datasets.

---

> > ### Comment · Reviewer_Jyb6 · 2021-11-27
> > **Thank you for your clear answer**
> >
> > After reading the other reviews and rebuttal, my opinion is unchanged and I will keep my score.

---

### Official Review · Reviewer_Kf27 · 2021-11-03

**Correctness:** 3
**Technical Novelty And Significance:** 2
**Empirical Novelty And Significance:** 2
**Recommendation:** 8
**Confidence:** 3

**Main Review:**

Strengths:

Empirical analysis, with theoretical justification, shows that SSL is more roboust than SL in feature learning from imbalance datasets.

Using a semi-synthetic dataset, they show that SSL learns label-irrelevant features that are useful for classifying rare classes.

Adapting SAM by reweighting more on low density instances improves performance.

Weaknesses:

Theorem 1 could have more explanation.

Table 1:  Gap Freq and Gap Rare were not described

Table 2;  Which of the four classes are frequent (or rare)?

**Summary Of The Paper:**

The authors explore the robustness of feature learning via self-supervised learning (SSL) and supervised learning (SL) with imbalance datasets.  Generally, SL can learn better features than SSL, and features are better from balanced than from imbalance datasets.  However, with imbalance data, SSL is more robust than SL in terms of the performance difference of features learned from balance and imbalance datasets.  That is, the performance does not drop as much for SSL with imbalance data from balanced data.   The robustness is observed from both in-domain (ID)  and out-of-domain (OOD) tasks (different downstream tasks).  They hypothesized that "SSL learns richer features from frequent data that are transferable to rare data."

Using a synthetic dataset with 2 frequent classes and 1 rare class, they observe that SL learns one feature to distinguish the two frequent classes and one feature to overfit the rare class.  However, SSL learns two features that can classify the 3 classes well. That is, SSL can learn label-irrelevant-but-transferable features from the frequent classes which can help classify the rare class.

To illustrate the observation from a synthetic dataset also exists in the real world, they construct a semi-synthetic dataset.  From CIFAR 10, they generate 5 fequent classes and 5 rare classes.  For the frequent classes, the right half of the image is replaced by a random half image, ie not relevant to the label.  For the rare classes, the left half is replaced by blank.  Using activation maps, they observe that SL learn features from only the left half (relevant to the label) but not from the right half (irrelevant to the label).   However, SSL learns features from both halves.

To further improve robustness, they adapt SAM to the imbalance scenario: rwSAM. SAM improves model generalization by penalizing loss sharpness. For SAM, the loss is uniformly low in the neighboring area.   To have a flatter region for rare examples, rwSAM reweights rare examples in the inner maximization step of SAM.  Since labels are not available, they estimate kernel density on feature vectors and the weight is inversely proportional to the density.  On two datasets, empirical results indicate rwSAM improves performance over SAM.



**Summary Of The Review:**

The authors propose well-designed experiments to show SSL is more robust than SL in feature learning from imbalance data and SSL learns label-irrelevant features that could be useful for the rare classes.   The presentation can be improved as indicated above.

---

> ### Author Response · Authors · 2021-11-20
> **Response to Reviewer Kf27**
>
> We thank the reviewer for the constructive comments, and for noting that our experiments are “well-designed”. We address the questions as below.
>
> **Q1**: Theorem 1 could have more explanation.
> **A1**: In the revision, we expanded the description of setting to emphasize the relationship between the theoretical results and empirical analysis and added a paragraph to clarify more on the insights of Theorem 1.
>
> **Q2**: In Table 1, Gap Freq and Gap Rare were not described.
> **A2**: Gap Freq and Gap Rare are described on $\underline{\text{page 8}}$: “Indeed, as shown in Table 1, we still observe a similar phenomenon—the frequent classes have much smaller pre-training generalization gap than the rare classes (0.035 vs. 0.081)” “Note that compared to SimSiam, rwSAM closes the generalization gap of pre-training loss on rare examples from 0.081 to 0.066, which verifies the effect of re-weighted regularization.” Gap Freq. refers to the gap between training and test SimSiam loss on the 5 frequent classes, while Gap Rare refers to that on the 5 rare classes. We defined Gap Freq and Gap Rare in detail and added a pointer in the revision.
>
> **Q3**: In Table 2, which of the four classes are frequent (or rare)?
> **A3**: CUB, Cars, Aircrafts, and Pets are four different **downstream tasks**, but not 4 classes. As we described the settings of the OOD evaluation in $\underline{\text{Section 2}}$, only the pre-training dataset has imbalance.

---

> > ### Comment · Reviewer_Kf27 · 2021-11-24
> > **comment on author response**
> >
> > Q&A1: Thanks for the added intuition, though it seems I have read it before.  Why do W_SL and W_SSL have different number of w's: 2 vs 3?   Does the analysis change if they have the same number of w's for a "fair comparison"?
> >
> > Q&A2:  Thanks for the description of Gap Freq. and Gap Rare in the paper.
> >
> > Q&A3:  I missed that.  Maybe adding a few words about the downstream tasks from Section 2 in the paragraph discussing Table 2 can help remind readers like me.
> >
> > Tables 1 and 2: looks like the improvement is within the error bars, hence not significant improvement (thanks for including the error bars).

---

> > > ### Author Response · Authors · 2021-11-25
> > > **Response to the comment**
> > >
> > > Thanks for providing the feedback! We will descrie the settings and the items in the tables more clearly in the future version.
> > >
> > >
> > >
> > > **Q1&A1:** The analysis will not change if we use more number of neurons ($w$’s) in either $W_{\\text{SL}}$ and $W_{\\text{SSL}}$. As long as $m \\ge 3$ for SL and $m \\ge 2$ for SSL, the proof and the conclusion is the same as the current case.
> > >
> > > For self-supervised learning, we proved in Theorem 3.1 (page 19) that the learned features $w_1, w_2$ are exactly the top 2 eigenvectors of the second moment $M$, and the interesting direction $e_2$ has a large correlation with their span. Our statement is still true for larger feature dimension $m>2$. This is because the learned features $w_1, w_2, \cdots, w_m$ now become the top $m$ eigenvectors of $M$, and the norm of the projection of $e_2$ onto their span will be even larger.
> > >
> > > For supervised learning, we proved in Theorem 3.1 that the interesting direction $e_2$ has small correlation with the span of learned features when the model is a 1-layer linear model. (The feature dimension is 3 because it’s a 3-way classification problem.) As we stated in the submission, any 2-layer network with intermediate dimension $m \\ge 3$ learns the same features as the 1-layer model. **This is intuitively because the supervised learning models is not incentivized to learn any features that are are more diverse than what are needed to predict labels.** Three dimensional features suffice to predict the label, and therefore SL won’t learn more than 3-dimensional features even the architecture has more than 3 neurons. Please see a more detailed proof sketch below for this fact. Therefore, our statement that $e_2$ has small correlation with the supervised features is still true for any $m \\ge 3$. We will also incorporate and expand this part in the future version to make the theoretical analysis clearer.
> > >
> > > **Tables 1 and 2:** We admit that the absolute improvement over SSL + SAM without re-weighting is not large. However, the main point of our paper is discovering that SSL is more robust to class imbalance than SL. Note that even SSL with balanced datasets is not better than SSL with imbalanced datasets significnatly, so we should not expect very large improvements. In addition, the improvement of SSL + rwSAM over the original SSL (without SAM) is significant in most cases.
> > >
> > > **Proof sketch of the 2-layer SL with $m\\ge3$**:
> > > Consider the case with one intermediate layer of dimension $m$, i.e. $f_{W_1, W_2}(x) \triangleq W_2W_1x$. We are looking for the minimal norm solution $\\|W_1\\|_F^2 + \\|W_2\\|_F^2$ subject to the margin constraint $f\_{W_1, W_2}(x)\_y\\ge f\_{W_1, W_2}(x)\_{y’}+1$ for all data $(x,y)$ in the dataset and $y’ \\ne y$. Note that $\\|W_1\\|_F^2 + \\|W_2\\|_F^2$ can be decomposed:
> > > $$\\|W_1\\|\_F^2 + \\|W_2\\|\_F^2 = \\sum\_{i=1}^m W\_{2,[1,i]}^2 + W\_{2,[2,i]}^2 + W\_{2,[3,i]}^2 + \\|W\_{1,[I]}\\|\_2^2
> > > \\ge \\sum\_{i=1}^m 2 \\sqrt{\\sum\_{j=1}^3 \\|W\_{2,[j,i]}W\_{1,[i]}\\|\_2^2} ,$$
> > > where the last line follows from the AM-GM inequality, and the equality holds iff $W\_{2,[1,i]}^2 + W\_{2,[2,i]}^2 + W\_{2,[3,i]}^2 = \\|W\_{1,[I]}\\|\_2^2$ for each $i \\in [m]$. Consider two sets of $W\_1, W\_2$ and $\\tilde{W\_1},\\tilde{W\_2}$ with $W\_{2,[j,i]}W\_{1,[i]} = \\tilde W\_{2,[j,i]}\\tilde W\_{1,[i]}$ and $\\tilde W\_{2,[1,i]}^2 + \\tilde W\_{2,[2,i]}^2 + \\tilde W\_{2,[3,i]}^2 = \\|\\tilde W\_{1,[i]}\\|\_2^2$. We know $f\_{W\_1, W\_2}(x) = f\_{\\tilde W\_1, \\tilde W\_2}(x)$ for each $x$, and $\\|W\_1\\|\_F^2 + \\|W\_2\\|\_F^2 \\ge \\|\\tilde W\_1\\|\_F^2 + \\|\\tilde W\_2\\|\_F^2$ from the inequality above. Therefore, all the solutions to the minimal norm problem subject to the margin constraint must satisfy $W\_{2,[1,i]}^2 + W\_{2,[2,i]}^2 + W\_{2,[3,i]}^2 = \\|W\_{1,[i]}\\|\_2^2$.
> > >
> > > By Cauchy-Shwartz we also have
> > > $$\\sum\_{i=1}^m 2 \\sqrt{\\sum\_{j=1}^3\\|W\_{2,[j,i]}W\_{1,[i]}\\|\_2^2} \\ge 2\\sqrt{\\sum\_{j=1}^3\\left\\|\\sum\_{i=1}^mW\_{2,[j,i]}W\_{1,[i]}\\right\\|\_2^2}.$$
> > > Therefore, all the solutions to the minimal norm problem subject to the margin constraint must make the equality holds. Consider $w\_1, w\_2, w\_3$, where $w\_j = \\sum\_{i=1}^mW\_{2,[j,i]}W\_{1,[i]}$ for $j \\in [3]$. Hence, $f\_{W\_1, W\_2}(x) = [w\_1,w\_2,w\_3]^\\top x$ for all $x$. Note that $\\|W\_1\\|\_F^2 + \\|W\_2\\|\_F^2 \\ge 2\\sqrt{\\|w\_1\\|\_2^2 + \\|w\_2\\|\_2^2 + \\|w\_3\\|\_2^2}$ by the inequality, and the equality holds when $w\_1 = W\_{2,[1,1]}W\_{1,[1]}, w\_2 = W\_{2,[2,2]}W\_{1,[2]}, w\_3 = W\_{2,[3,3]}W\_{1,[3]}$ and all other entries in $W\_2$ and $W\_1$ are 0. Therefore, the original minimal norm problem subject to the margin constraint can be reduces to the one-layer case, i.e. minimize $\\|w\_1\\|\_2^2 + \\|w\_2\\|\_2^2 + \\|w\_3\\|\_2^2$ subject to the margin constraint $w\_y^\\top x \\ge w\_{y’}^\\top x + 1$ for all empirical data $x$.

---

### Decision · Program_Chairs · 2022-01-20

**Decision:**

Accept (Spotlight)

**Comment:**

This paper is proposed to investigate the robustness of self-supervised learning (SSL) and supervised learning (SL) in both balanced (in domain) and imbalanced (out of domain) settings. It can be concluded that SL can regularly learn better representations than SSL, and representations are better from balanced than from imbalanced datasets. The SSL is more robust than SL in the imbalanced settings, which is the crucial of this paper. Expect the experimental results, the authors also provided theoretical analysis to support their claims. The authors also extend a well-established method SAM into the Reweighted SAM as the technical contribution to better address the imbalanced setting. The paper is well written with clear logic to follow.